# The Phenolics and Antioxidant Properties of Black and Purple versus White Eggplant Cultivars

**DOI:** 10.3390/molecules27082410

**Published:** 2022-04-08

**Authors:** Nesrin Colak, Aynur Kurt-Celebi, Jiri Gruz, Miroslav Strnad, Sema Hayirlioglu-Ayaz, Myoung-Gun Choung, Tuba Esatbeyoglu, Faik Ahmet Ayaz

**Affiliations:** 1Department of Biology, Faculty of Science, Karadeniz Technical University, Trabzon 61080, Turkey; ncolak@ktu.edu.tr (N.C.); aynur-leyla@hotmail.com (A.K.-C.); sha@ktu.edu.tr (S.H.-A.); 2Department of Experimental Biology, Palacky University Olomouc, Slechtitelu 27, 783 71 Olomouc, Czech Republic; jiri.gruz@upol.cz; 3Laboratory of Growth Regulators, Faculty of Science, Palacky University & Institute of Experimental Botany AS CR, Slechtitelu 27, 783 71 Olomouc, Czech Republic; miroslav.strnad@upol.cz; 4Department of Herbal Medicine Resource, Dogye Campus, Kangwon National University, Hwangjori 3, Dogye-up, Samcheok 25949, Korea; cmg7004@kangwon.ac.kr; 5Institute of Food Science and Human Nutrition, Gottfried Wilhelm Leibniz University Hannover, Am Kleinen Felde 30, 30167 Hannover, Germany

**Keywords:** aubergine, *Solanum melongena*, fruit, anthocyanin, phenolic acid, antioxidant capacity, α-amylase, diabetes

## Abstract

The total phenolic content, anthocyanins, phenolic acids, antioxidant capacity and α-amylase inhibitory activity of black (Aydin Siyahi), purple (Kadife Kemer) and white (Trabzon Kadife) eggplants grown in Turkey were subjected to a comparative investigation. The black cultivar exhibited the highest total phenolic (17,193 and 6552 mg gallic acid equivalent/kg fw), flavonoid (3019 and 1160 quercetin equivalent/kg fw) and anthocyanin (1686 and 6167 g delphinidin-3-*O*-glucoside equivalent/kg fw) contents in crude extracts of the peel and pulp. The majority of the caffeic acid was identified in the ester (2830 mg/kg fw) and ester-bound (2594 mg/kg fw) forms in the peel of ‘Kadife Kemer’ and in the glycoside form (611.9 mg/kg fw) in ‘Aydin Siyahi’, as well as in the pulp of these two eggplants. ‘Kadife Kemer’ (purple eggplant) contained the majority of the chlorogenic acid in free form (27.55 mg/kg fw), compared to ‘Aydin Siyahi’ in the ester (7.82 mg/kg fw), glycoside (294.1 mg/kg dw) and ester-bound (2.41 mg/kg fw) forms. The eggplant cultivars (peel and pulp, mg/kg fw) exhibited a relatively high delphinidin-3-*O*-rutinoside concentration in the peel of ‘Aydin Siyahi’ (avg. 1162), followed by ‘Kadife Kemer’ (avg. 336.6), and ‘Trabzon Kadife’ (avg. 215.1). The crude phenolic extracts of the eggplants exhibited the highest antioxidant capacity values (peel and pulp, µmoL Trolox equivalent/kg fw) of 2,2-diphenyl-1-picrylhydrazyl (DPPH, 8156 and 2335) and oxygen radical absorbance capacity (ORAC, 37,887 and 17,648). The overall results indicate that black and purple eggplants are the cultivars with greater potential benefits in terms of their phenolics and antioxidant values than the white eggplant.

## 1. Introduction

As potent phenolic antioxidants with over 8000 identified compounds, polyphenols are present at high concentrations in a variety of fruits and vegetables. These have been the subject of considerable attention in recent years due to their vital roles in health maintenance through the regulation of metabolism, weight, and chronic disease, cell proliferation, and reduction in the risk of coronary heart diseases, neurodegenerative diseases, and certain forms of cancer [1,2]. 

Ranking among the top 10 in terms of oxygen radical absorbing capacity (ORAC) due to the fruits’ polyphenol constituents, the cultivated eggplant (*Solanum melongena* L, Fam: Solanaceae) also known as aubergine, brinjal, melanzane, berenjena or patlıcan is an economically important vegetable crop of tropical and subtropical zones of the world including Asia (47,142,210 tons), Africa (1,814,535 tons), Europe (936,642 tons), the Americas (295,387 tons) and Oceania (4342 tons) [3,4,5,6,7]. Eggplant is grown over 1.7 million hectares world-wide. Turkey ranks fourth in the world in eggplant production with an annual production of approximately 816,000 tons, or 2% of world production [8,9]. 

Eggplant exhibits wide diversity in color (purple, purple/black, green, or variegated), size (ave. 210.4 g, range 42–464) and shape (elongated, ovoid, or slender). Based on fruit shape, eggplant is classified as egg-shaped (*S. melongena* var. *esculentum*), long and slender in shape (*S. melongena* var. *serpentinum*), or dwarf types (*S. melongena* var. *depressum*). The oblong to elongate-shaped dark purple, purple/black or violet eggplants are used worldwide, but other varieties that differ in color, size and shape are also known [3,4,6,7,10,11,12].

Fruit and vegetables are food sources of a variety of non-nutritive bioactive compounds (mainly phenolics), their long-term or regular consumption of which is associated with a reduced risk of certain types of cancers, and cardiovascular and other neurodegenerative diseases [1,13,14]. Polyphenols are widely distributed in nature; approximately 8000 different structures of these plant compounds have been identified to date. The most common classification of phenolic metabolites are based on two aromatic rings connected by a bridge consisting of three carbons (C_6_-C_3_-C_6_), which distinguishes the flavonoid and non-flavonoid compounds. Flavonoids consist of more than 6000 types and are divided into six different main subclasses (anthocyanins, chalcones, flavanones, flavones, flavonols and isoflavones), depending on the carbon of the C-ring on which the B ring is attached and the degree of unsaturation and oxidation of the C-ring. The physiological state flavonoids usually occur in association with sugar as glycosides. The second class of plant phenolics -non-flavonoid metabolites-consists of the following subgroups: phenolic acids (hydroxybenzoates C_6_-C_1_, hydroxycinnamates C_6_-C_3_), lignans (C_6_-C_3_)_2_ and stilbenes C_6_-C_2_-C_6_. Two other subclasses of non-flavonoid compounds are tannins (hydrolysable and condensed) and lignins; these have high molecular weights that lack a defined primary carbon base and occur mainly as complicated biopolymers, being unique for the particular polyphenols [14,15,16]. 

The major phenolic compounds in eggplant fruits are reported to be highly beneficial for human health due to their known biological activities, and they can be potentially used in the treatment of several metabolic and cardiovascular diseases [17]. Delphinidin derivatives are the only anthocyanins identified in eggplant fruits. The most common anthocyanin structure in the peel of eggplant fruits is delphinidin-3-(*p*-coumaroyl-rutinoside)-5-glucoside, known as nasunin [18], while the main phenolic acid in the flesh is chlorogenic acid (CGA), together with its hydroxycinnamic acid conjugates (chlorogenic acid isomers, isochlorogenic acid isomers, amide conjugates, unidentified caffeic acid conjugates, and acetylated chlorogenic acid isomers) varying from 75 to 94% of the total phenolic content in a wide range of eggplant cultivars [3,4,6,8,9,10,18,19]. 

Studies have indicated that anthocyanins and phenolic acids contribute to high antioxidant properties in eggplant [17,18]. Eggplant exhibits potential health benefits in a number of degenerative diseases, cancer, cardiovascular diseases, diabetes, pulmonary disorders, and Alzheimer’s disease [6,12,20,21,22,23]. In addition, studies have shown that eggplant varieties enriched in phenolic phytochemicals and moderate free radical scavenging-linked antioxidant activity have a potential to reduce hyperglycemia-induced vascular complications resulting from oxidative damage [20,24,25,26]. In this respect, a previous in vitro study showed that several eggplant samples (e.g., White pulp and Graffiti skin) had high α-glucosidase inhibitory activity combined with low α-amylase inhibitory activity, indicating a potential ability to reduce glucose absorption in the intestine [20].

The three cultivars in the present study are classified in terms of their characteristic color: dark black/purple in the case of ‘Aydin Siyahi’, purple with white stripes or mottles in ‘Kadife Kemer’, and occasional light purple stripes in the mostly white ‘Trabzon Kadife’ [27]. The white cultivar (Trabzon Kadife) has recently been described as a new cultivar, and its polyphenol oxidase activity during fruit maturation has been well-documented [27]. This eggplant is grown in the Black Sea region of Turkey and enjoys increasing market value. However, information about the phenolics and antioxidant capacity of the new cultivar as well the other two, black and purple, cultivars is scant and not well-documented, particularly in terms of their antioxidant and enzyme inhibitory activities (α-amylase). The present study examined the phenolic content, antioxidant capacity and the distribution of hydroxybenzoic and hydroxycinnamic acid derivatives in the peel and pulp of the three eggplant cultivars from Turkey for the first time.

## 2. Results and Discussion

### 2.1. Variation in Total Phenolic Compounds (TPC), Flavonoid (TF) Contents and Antioxidant Capacity (AC) Values in Eggplants

The TPC, TF and ACY contents in the peel and pulp extracts of the studied eggplants are summarized in Table 1. The contents varied significantly (*p* < 0.05) in terms of the crude extracts (CE) and their three further fractions, aqueous (AF), polyphenolic (PPF) and anthocyanin (ACYF), depending on the peel color. The phenolic contents and AC values of the peels and pulps were statistically strongly correlated among the eggplants (r = 0.956–0.997, *p* < 0.05) or crude extract and its further three fractions (r = 0.830–1.000, *p* < 0.05), except for the TF content of the aqueous phase. In the present study, the black eggplant, ‘Aydin Siyahi’, exhibited the highest TPC content in the crude extract (CE) of the peel (17,193 mg gallic acid (GA) equivalent (E)/kg fw) and pulp (6551 mg GAE/kg fw), followed by their aqueous (AF), polyphenolic (PPF) and anthocyanin (ACYF) fractions. Similarly, the black eggplant (Aydin Siyahi) exhibited the highest TF content in the peel (3019 mg quercetin equivalent (QE)/kg fw) and pulp (1159 mg QE/kg fw), followed by their three subextracts/fractions (Table 1). Reports have also shown a wide range in TPC (range; 22–20,490 mg gallic acid, chlorogenic acid or caffeic acid equivalent/kg fw) and TF (range; 30–39,540 mg catechin or quercetin equivalent/kg fw) contents in the peel of eggplants [3,28,29,30]. Our results also agree with these reported ranges for TPC and TF contents.

The total ACY content in the eggplants was also cultivar-specific and varied significantly (*p* < 0.05) among the cultivars. Peel from the black eggplant (Aydin Siyahi) exhibited the highest total ACY content (16,836 del-3-glc equivalent g/kg fw) followed in descending order by ‘Kadife Kemer’ and ‘Trabzon Kadife’ (12,081 and 8487 g del-3-glc equivalent/kg fw, respectively) (Table 1). The results of the current study are in close agreement with the reported values for ACY content in the literature (between 90 and 19,750 mg/kg dw vs fw) [3]. 

The AC values in the eggplant extracts/subextracts (e.g., fractions) were determined using DPPH and ORAC assays (Table 1). The peel of the black eggplant exhibited the highest DPPH and ORAC values for CE (8156 and 37,886, µmoL Trolox equivalent (TE) kg^−1^ fw) and ACYF (7646 and 31,929 µmoL TE kg^−1^ fw), followed by the purple (3366 and 22,670 µmoL TE kg^−1^ fw) and white (2634 and 19,554 µmoL TE kg^−1^ fw) eggplants. Similarly, the pulp of the black eggplant exhibited the highest DPPH and ORAC values for CE (2334 and 17,648 µmoL TE kg^−1^ fw) and ACYF (1872 and 9718 µmoL TE kg^−1^ fw), followed by the purple and white eggplants. In addition, the AC values (DPPH and ORAC, Table 1) contents were significantly (*p* < 0.05) strongly correlated with the TPC and TF contents (range; r = 0.934–1.000, *p* < 0.05) within the peel and pulp or among the eggplants. These values indicate that the AC is strongly related to the TPC and TF or ACY contents in the eggplants. The ACYF, followed by PPF, exhibited the highest TPC, TF and ACY contents and also appeared to be the major contributor to the antioxidant capacity, while the AF exhibited the lowest contents. In general, based on the correlation matrix, the TPC and TF contents in the AF did not correlate with the AC capacity values, in most cases exhibiting insignificant negative low and high correlations (data not shown). These results are in good accordance with other reports [3], and confirm that eggplant peel—black, purple, or white—constitutes an outstanding source of phenolics with high AC. 

### 2.2. Variation in Phenolic Acids in Eggplants

Phenolic acids identified and quantified in free (F), ester (E), glycoside (G) and ester-bound (EB) forms of phenolic acids in the eggplants were cultivar-specific, and their concentrations (mg/kg fw) in the peel and pulp differed significantly (*p* < 0.05) (Table 2). Caffeic acid (CaA), a hydroxycinnamic acid (HCA) derivative, exhibited the highest concentration in the peel in free (3.77), ester (2830) and ester-bound (2594) forms in ‘Kadife Kemer’, and in glycoside form (611.9 mg/kg fw) in ‘Aydin Siyahi’. In the case of the pulp, CaA was also the major acid, and its concentrations were high in the black eggplant (Aydin Siyahi) in free (0.61), ester (2512), glycoside (195.0) and ester-bound (1824) forms. The white-colored eggplant ‘Trabzon Kadife’ contained considerable amounts of CaA in the peel (2316) and pulp (2457.75) in ester and ester-bound (2067.31 and 1122.41) forms. Ferulic acid (FeA) content was also cultivar-specific, with a significantly higher concentration (*p* < 0.05) in the peel and pulp of ‘Aydin Siyahi’ in ester form (243.36 and 341.41), of ‘Kadife Kemer’ and ‘Trabzon Kadife’ in glycoside form (48.02 and 59.86 and in the black eggplant in ester-bound form (335.09 and 265.41), respectively (Table 2).

*p*-Hydroxybenzoic acid (*p*-HBA), gallic acid (GaA), syringic acid (SyA), protocatechuic acid (PCA) and vanillic acid (VaA) are common hydroxybenzoic acid derivatives (HBAs) and are usually present in bound form in foods [31]. The eggplants in the present study contained considerable amounts (mg/kg fw) of these phenolic acids. The second most abundant phenolic acid in free form in the peel as HBAs was *p*-HBA, the highest level of which was observed in ‘Kadife Kemer’ (5.57), followed by ‘Trabzon Kadife’ (5.00), and finally ‘Aydin Siyahi’ (2.15). The black eggplant exhibited the highest contents of gallic acid (GaA), protocatechuic acid (PCA) and *p*-HBA in ester (458.47, 50.42 and 30.37, respectively), glycoside (399.07, 43.60 and 85.19, respectively) and ester-bound (181.16, 39.09 and 41.51, respectively) forms in the peel. In the pulp, the first two acids (GaA and PCA) exhibited the highest contents in ester form (144.81 and 35.40), followed by *p*-HBA and PCA contents in glycoside (51.75 and 38.91), and GaA and PCA contents in ester-bound (19.47 and 28.99) forms (Table 2). 

The most widely distributed phenolic acid ester in eggplants is chlorogenic acid (CGA, 5-caffeoylquinic acid), and these vegetables are highly prized in the human diet because of this high content [3,18,32]. In the present study, CGA was also detected in four forms (Table 3), its concentration (mg/kg fw) in peel being significantly (*p* < 0.05) higher than that in the pulp. For instance, CGA occurred at high concentrations in free form in the peel of ‘Kadife Kemer’ (27.55), in the peel of the black eggplant (Aydin Siyahi) in ester (7.82), glycoside (294.1) and ester-bound (2.41) forms. However, the pulp contained the highest CGA concentration in free (4.71), ester (3.06) and ester-bound (1.23) forms in Aydin Siyahi, and in the glycoside form (60.12) in ‘Kadife Kemer’. Our findings are in close agreement with those of other authors who reported a broad variation in CGA concentration in eggplants, ranging from 8600 to 17,000 mg/kg fw [3,32].

### 2.3. Variation in Anthocyanin Composition in Eggplants

Figure 1 shows the presence of detected anthocyanins in the eggplants. The major anthocyanin in eggplant in the present study was delphinidin-3-*O*-rutinoside (del-3-rut) (Table 4). The LC-MS data in the present study also confirmed the presence of del-3-rut-5-glc, del-3-rut-glc and del-3-glc, although there were also trace anthocyanins. The eggplants exhibited significantly higher del-3-rut concentrations in the peel of ‘Aydin Siyahi’ (avg. 1162.22 mg/g fw), followed by ‘Kadife Kemer’ (avg. 336.59 mg/kg fw) and ‘Trabzon Kadife’ (avg. 215.11 mg/kg fw), in comparison to the pulp (range: 45.45–194.62 mg/kg fw). 

Nasunin, a delphinidin derivative (delphinidin-3-(*p*-coumaroylrutinoside)-5-glucoside) first reported in Japanese eggplants, is the major anthocyanin found in eggplant peel [3,23]. These anthocyanins have also been reported as the major anthocyanins in different varieties of eggplants reported from Bulgaria [33], Japan [34] and the United States [35]. Sadilova et al. [10] detected the same pattern of delphinidins (delphinidin-3-rutinoside, delphinidin-3-rutinoside-5-glucoside, delphinidin-3-glucoside) in eggplants to that earlier determined by Wu and Prior [35] in eggplants reported from the USA. In a recent study, Calumpang et al. [36] also noticed the presence of del-3-rut (described as anthocyanin_I, delphinidin-3-*O*-(-feruloyl) rutinoside) in a purple Japanese (Takii) eggplant ‘Ryoma’ (*S. melongena*). Acylated anthocyanins (*p*-coumaroyl, feruloyl or caffeoyl acyl moieties) are the most abundant forms in eggplants, although some accessions are found in the latter, in which a non-acylated anthocyanin, namely del-3-rut, is predominant [10,34,37]. However, except for del-3-rut, non-acylated anthocyanins account for only a small proportion of the total anthocyanin content. Despite the general structural similarity of anthocyanins in eggplants, deviations can sometimes be observed [18]. For instance, in the peel of the ‘Zi Chang’ eggplant, only two anthocyanins, delphinidin-3-glucoside-5-(coumaryl)-dirhamnoside and delphinidin-3-glucoside-5-dirhamnoside, are found in position 3, which carries a single glucose moiety instead of the common *p*-coumaroyl-rutinoside, while position 5 is conjugated with a dirhamnosyl moiety [38]. This suggests the existence of a genetic variation for enzymes such as glycosyltransferases, which mediate the conjugation of anthocyanidins with sugar moieties [18,38]. Anthocyanins are reported to be involved in pigmentation, specifically purple to black pigmentation in the peel of eggplant fruit [34,36]. 

In the present study, we compared anthocyanins in the peel of black, purple, and white eggplants. Discoloration and color-changing phenomena have been observed in plant tissues during development. Anthocyanin discoloration may be due to either anthocyanin reduction in plant tissues or to structural changes in the anthocyanin that leads to a loss of color that is controlled by active enzyme-driven breakdown processes (e.g., polyphenol oxidase (PPO), peroxidase (POD) and β-glucosidases) or non-enzymatic factors-attributed to either reduced biosynthesis or increased degradation of anthocyanins, or a combination of both. In the anthocyanin biosynthetic pathway, the expression of late biosynthetic genes ((LBGs—*F3*′*H*, *F3*′*5*′*H*, *DFR*, *ANS*, and *UFGT*)) are required for the biosynthesis of specific classes of flavonoids, including anthocyanins, and determines the quantitative variation in anthocyanins. Positive correlations between the expression levels of LBGs and the anthocyanin content have been consistently observed in many Solanaceous vegetables, including eggplant [18]. Transcript levels of late biosynthetic genes decrease during the later stages of ripening when discoloration occurs. Anthocyanin biosynthesis is regulated by MBW complexes consisting of different MYBs, but with the same bHLH and WD40 transcription factors. Reduced biosynthesis is controlled by the downregulation of MYB activators and the upregulation of MYB repressors. The expression level of *SmCHS* in eggplant has been reported to be significantly upregulated in black (Black Beauty) or violet (Classic) fruits compared to the green (genotype E13GB42) or white (Ghostbuster) mutants [39,40]. In addition, the transcript levels of *SmCHS* and *SmCHI*, but not *SmF3H*, have been shown to correlate well with the anthocyanin accumulation pattern in the eggplant ‘Lanshan Hexian’ [41]. Studies have also emphasized that non-enzymatic factors also have a considerable effect on the chemical structure of the anthocyanins that determine anthocyanin color and stability and may enhance the vulnerability of the enzymes that degrade anthocyanins. The higher the level of B-ring hydroxylation, the more purple the color, but the more unstable the anthocyanins are [42]. The effect of glycosylation varies depending on the number and the position of the sugar moieties [43]. In addition, glycosylation at C3 elevates stability and shifts the color slightly toward red. The stabilizing effect of diglycosides at C3 is stronger than that of monoglycosides. In contrast, glycosylation at C5 reduces pigment intensity. Acylation increases anthocyanin stability, and an increasing number of acyl moieties causes a color shift from red to blue [43,44]. 

### 2.4. Variation in α-Amylase Activity in Eggplants

α-Amylase inhibition activity was highest in the peel (66.37%) and pulp (85.03%) from ‘Kadife Kemer’ but much lower in ‘Trabzon Kadife’ (45.93 and 62.35%, respectively) and ‘Aydin Siyahi’ (37.73 and 57.49%) (Figure 2) in the present study. This assumed that the phenolic composition of the pulps exhibited higher inhibitory activities than that in the peel. Previous research with selected food extracts reported an association between antioxidant activity and α-amylase inhibition activity [20]. The purple eggplant (Kadife Kemer) exhibited moderate AC values determined by the DPPH and ORAC assays, although it had the highest α-amylase inhibition activity. In a much earlier study, Kwon et al. [20] compared the α-amylase inhibition activity of phenolics in the peel and pulp of ‘Purple’, ‘White’, ‘Graffiti’ and ‘Italian’ eggplant cultivars. They reported moderate low α-amylase inhibitory activities in these four eggplants combined with moderate AC values [20]. Similarly, the white-coloured eggplant (Trabzon Kadife) in the present study exhibited comparable α-amylase inhibitory activities in comparison to the black eggplant (Aydin Siyahi). 

Consistent with this, the difference in α-amylase activity inhibition in the eggplants can be attributed to the above-cited chemical (different phenolic molecular structure, anthocyanin or non-anthocyanin phenolic compounds, pH, PPO activity, etc.) and geographical factors [20,21,22,23,24,25,26]. Numerous studies have shown that polyphenol oxidase (PPO) activity varies among eggplants, and also that some varieties exhibit high phenolic content and low browning capacity. Factors such as the intracellular pH, which affects the activity of PPO, or the presence of ascorbic acid in the fruit flesh tissues, which prevents the oxidation of *ortho*-diphenols, may also play a role in the modification of the browning process in eggplants, with both factors, therefore, affecting the phenolic constituents. Studies have indicated no correlation with either the degree of browning or the color difference in eggplants. Other factors, such as different PPO activities among different varieties or other cellular factors, such as the size of cells and interstitial spaces, which may differ among different varieties of a given species, may play a role in the browning and color evolution of the fruit flesh. In an earlier report, concerning a detailed PPO characterization in ‘Trabzon Kadife’, Torun et al. [27] reported that white eggplant had a fast browning capacity and low ascorbic acid content, and all of these factors can, therefore, induce the oxidation of *ortho*-diphenols exhibiting different phenolic status among eggplants. 

The interaction between plant polyphenols and α-amylase activity inhibition has become the subject of recent interest in postprandial hyperglycemia [25]. Accordingly, the consumption of starch largely determines postprandial blood sugar levels, and also affects glucose metabolism [25]. Postprandial hyperglycemia has been implicated in the disturbance of carbohydrate metabolism. Delaying any increase in blood glucose levels is, therefore, regarded as useful for the mitigation of insulin resistance and/or type II diabetes. Starch is largely digested to reducing sugars (such as maltose, maltotriose and amylodextrin) by α-amylase in the small bowel. These reducing sugars are, subsequently, further hydrolyzed by α-glucosidase, resulting in glucose. α-Amylase is, therefore, a particularly important enzyme in starch hydrolysis. Studies have recommended that enzyme activity be regulated by both chemical and biological components in order to prevent and treat postprandial hyperglycemia and associated metabolic disorder [24,25]. There is a very close association between the inhibitory activity of a polyphenol against α-amylase and the phenolic molecular structure, and the relationships between structure and inhibition have been the subject of previous investigation [25]. In terms of flavonoids, in particular, the presence of hydroxyls (–OH) at the 5-, 6-, and 7-positions of ring A and at the 4′-position of ring B is capable of increasing the inhibitory activity due to the important role played by –OH in the formation of hydrogen bonds with the enzyme’s active site [45]. The conjugation of 4-carbonyl with 2, 3-double bonds also makes a significant contribution to the flavonoids’ inhibitory properties. This is principally due to this conjugation heightening electron delocalization between the A- and C-rings, thus enhancing the stability of π-π stacking between the flavonoid aromatic rings and the indole ring of tryptophan at the active site of α-amylase [25,45]. Moreover, galloyl moiety has recently been proposed as an essential substitution for α-amylase inhibition by tea polyphenols and gallotannins [25,26]. This is attributable to the relatively powerful non-covalent interactions taking place between the moiety and the enzyme, including the hydrogen bondings between -OH of galloyl and the catalytic amino acid residues (e.g., Glu^233^), and hydrophobic π-π conjugation (aromatic–aromatic) between the galloyl benzene ring and tryptophan aromatic rings at the enzyme active site [25]. It is generally agreed that the antioxidant activity of phenolic compounds is often related to the chemical composition of individual compounds, which is dependent on a variety of factors, such as geographic variation, harvest time, environmental and agronomic conditions, the botanical parts of plants, and extraction methods [46]. 

### 2.5. Principal Component Analysis (PCA)

PCA is the most popular multivariate statistical analysis used by almost all scientific disciplines. It analyzes a data table representing observations described by several dependent variables, which are, in general, inter-correlated. Its aim is to extract the important information from the data table and to express this information as a set of new orthogonal variables known as principal components. It also represents the patterns of similarity of observations and variables by displaying them as points in maps [47] that can summarize the dimensionality of high-dimensional complex data through a smaller set of “summary indices” that can be easily visualized and analyzed. In recent years, researchers have used this method of analysis to determine whether any of the observations or variables differ significantly among treatments [48,49,50]. In the present study, the PCA was carried out to study the variation in contents of the total phenolics, phenolic acids and chlorogenic acids liberated in four forms, the anthocyanin/s (del-3-rut), and the antioxidant capacity values in the peel and pulp of three eggplants (black, purple, and white). The PCA showed that all of the chemical components determined in the eggplants were closely associated and significantly (*p* < 0.05) positively and strongly correlated with the peel and pulp, showing a total variance ranged between 81.56 and 99.93%. Accordingly, two principal components, explaining 99.69% of the overall variance (PC1; 98.61% and PC2; 1.08%), divided the pulp and peel in conjunction with the ORAC values and TPCs (Figure 3A). Noticeably, the PC1 (98.61%) is clearly identified with the pulp and closely associated with the ORAC values and the phenolic contents (upper positive side). However, the TPC and TF contents of the ACYF in the eggplant peels were closely associated with the ORAC on the PC1 (lower positive side). The major factor scores contributing to the PC1 (positive side) were CE/ACYF-TPC/TF-ORAC (5.947 and 4.155) (Table 5). In contrast, the main contributors to the PC2 (the negative side) were the AF and PPF in relation to the AC, but these were very low, and the data are not shown. The common feature of ‘Aydin Siyahi’, ‘Kadife Kemer’ and ‘Trabzon Kadife’ was thus the highest content of polyphenolic compounds in the CE and ACYF in the peel and the ORAC values. 

The phenolic acids in free (F) form in the peel and pulp of ‘Trabzon Kadife’, pulp of ‘Kadife Kemer’ in the upper quadrant and the peel and pulp of ‘Aydin Siyahi’ and ‘Kadife Kemer’ (pulp alone) in the lower quadrant on PC1 (60.48%, upper positive side) were closely associated with *p*-HBA and CGA (Figure 3B). Based on the correlation matrix, the free phenolic acid contents were significantly and strongly correlated with the peel and pulp of the studied eggplants (r = 0.873–0.990, *p* < 0.05). The remaining seven phenolic acids were located on the negative side on PC (37.47% variance). Among the five component scores, F1 exhibited the largest positive association with CGA (4.215) and *p*-HBA (3.454 and 3.452 with F2) (Table 5). 

Ferulic acid and CaA in the ester form (E) were closely associated with the peel of ‘Trabzon Kadife’ and ‘Kadife Kemer’ and the pulp of ‘Aydin Siyahi’ and ‘Kadife Kemer’ on PC1 (82.63%, horizontal axis positive side) (Figure 3C). The pulp of ‘Trabzon Kadife’ was located at the upper quadrant on PC1, close to the vertical axis (positive side). The remaining phenolic acids were located at the negative side on PC2 (16.67% variance, Figure 3C). The phenolic acids in ester form were also significantly correlated with the fruit parts (peel and pulp) in the eggplants r = 0.972–0.998 (*p* < 0.05), except for the pulp of ‘Trabzon Kadife’. In terms of the factor scores, F1 exhibited the largest positive associations with CaA (7.960) and PCA (3.594) (Table 5). 

The FeA, PCA, *p*-HBA, GaA and CaA in glycoside (G) form were closely associated with the eggplant peel and pulps on PC1 (52.74%, upper/lower quadrants, positive side) (Figure 3D). However, the remaining phenolic acids on PC2 (28.83%, upper/lower plans, positive side) were associated within, but not with, the peel and pulp. The phenolic acid contents in this form were significantly and strongly correlated with the peel and pulp in ‘Aydin Siyahi’, ‘Kadife Kemer’ and ‘Trabzon Kadife’, r = 0.825–0.922 (*p* < 0.05). Here, the largest positive or negative associations were attributed to F1 and F2 for CaA (4.258 and −3.325) and CGA (3.824 and 2.795) and F3 for FeA (2.210) (Table 5). 

FeA and CaA in the ester-bound (EB) forms were closely associated with the peel of ‘Aydin Siyahi’ and ‘Kadife Kemer’, the pulp of ‘Aydin Siyahi’, the pulp of ‘Trabzon Kadife’ and ‘Kadife Kemer’ and the peel of ‘Trabzon Kadife’ on the right on PC1, explaining 99.71% of the data variation. The remaining phenolic acids were associated within the peel and pulp only (Figure 3E). In contrast to the correlations of those above, three phenolic acid forms—the phenolic acids in EB form—were significantly and strongly correlated within the peel and pulp among the eggplants (r = 0.991–1.000, *p* < 0.05). The largest association belongs to F1 for CaA (8.055), while the remaining factor scores (F2–F5) have low associations (Table 5). Overall, the PCA was carried out separately for all forms of phenolic acids determined in the eggplant peel and pulp (Figure 3F). The PCA model accounted for 98.78% of the total variance (PC1, 82.20%; PC2, 16.58%). Again, the CaA-E and -EB forms were closely associated with the peel and pulp of the eggplants on PC1, except for the pulp of ‘Trabzon Kadife’, which closely associated with the PCA-E form. In addition, no correlation was found between the pulp of the white eggplant (Trabzon Kadife) and any forms of phenolic acids, while the peel and pulp of the remaining two eggplants were significantly and strongly correlated (r = 0.967–0.993, *p* < 0.05). The largest association belongs to F1 for CaA-E (11.632) and -EB (10.054), while the remaining factor scores have low associations (Table 5).

## 3. Materials and Methods 

### 3.1. Chemicals and Reagents

All of the solvents were analytical grade, of high-performance liquid chromatography-mass spectrometry (HPLC-MS) quality (>99%) and were obtained from Merck (Darmstadt, Germany). Deionized water (DIW) was used for the extractions and was of Milli-Q quality for liquid chromatography (LC) and UV-VIS spectrophotometric measurements. The delphinidin-3-*O*-rutinoside was purchased from Phytolab (Vestenbergsgreuth, Germany). The phenolic acid standards (protocatechuic acid, *p*-hydroxybenzoic acid, gallic acid, gentisic acid, vanillic acid, salicylic acid, syringic acid, caffeic acid, *p*-coumaric acid, *m*-coumaric acid, ferulic acid, *o*-coumaric acid and chlorogenic acid) were purchased from Sigma-Aldrich (Poole, Dorset, UK). The acarbose was purchased from J&K (Beijing, China), α-Amylase solution from Sigma Aldrich (Steinheim, Germany), 3,5-dinitrosalicylic acid (DNS) from Fluorochem (Hadfield, UK) and Na-K tartrate tetrahydrate (p. A.) from Chemsolute (Roskilde, Denmark). All of these pure standards were used for identification and quantification purposes.

### 3.2. Plant Material

Mature eggplant fruits (Figure 4) of commercial market size were obtained from local growers in Antalya and Mersin, Turkey, in the case of ‘Aydin Siyahi’ and ‘Kadife Kemer’, and from local growers in Giresun, Trabzon and Rize, Turkey, in the case of ‘Trabzon Kadife’. Eight eggplant fruits from six greenhouses for each cultivar were randomly selected. Plant or animal debris was immediately removed from the eggplant fruits, which were washed in double-distilled water, kept below 5 °C, and transported within approximately 3 h in a portable cold storage box. At the laboratory, the peel and pulp samples were prepared in line with the sampling protocol described by Stommel and Whitaker [4] for eggplants, with slight modifications. In brief, the fruits were peeled using a porcelain fruit/vegetable peeler within 1 h. A 2.5 cm longitudinal section from stem to blossom end was then collected from the middle of the fruit. The excised, deseeded tissue was immediately diced using a porcelain knife, frozen in liquid N_2_, and lyophilized (Christ, Alpha 1–2LD plus, Osterode, Germany). The dried peel and pulp samples from each eggplant were then pulverized using an agate mortar and pestle and stored at −80°C for further analyses. 

### 3.3. Extraction of Phenolics

Crude phenolic extracts (CE) of the peel and pulp were prepared by modifying the method described by Rodriguez-Saona and Wrolstad [51]. All of the extractions were performed in triplicate. Approximately 3 g of eggplant sample, prepared as described above, was extracted using 50 mL of 80% aqueous methanol (80:20, methanol:water, *v*/*v*), followed by triple extraction using the same solvent until a clear supernatant was obtained. The homogenates were combined and centrifuged at 1500 rpm in a M2 rotor (Hermle Z 326 K, Hermle Labortechnik, Wehingen, Tuttlingen, Germany) for 30 min at 4 °C. The supernatants were concentrated using a rotary evaporator (Laborata 4003, Heidolph Instruments, Schwabach, Germany) at 38 °C. The slurry was dried using a freeze-dryer and dissolved in 10 mL deionized water (aqueous extract) for further analysis. 

Next, the aqueous extract was fractioned via solid-phase extraction (SPE) using Thermo HyperSep™ C18 cartridges (max 500 mg packed bed, 3 mL, Waltham, MA USA) to obtain the subextracts (fractions). The extraction columns were rinsed with 80% methanol (9 mL) and then activated using deionized water (9 mL) followed by a triple wash. The aqueous sample was then passed through the columns.

Sugars and other polar compounds were first eluted (aqueous fraction) with deionized water, referred to as the aqueous fraction (AF). Next, ethyl acetate (9 mL) was passed through the columns to yield a polyphenolic fraction (PPF). Finally, 9 mL of acidified methanol (0.01% HCl) was employed for the third fraction, the anthocyanin fraction (ACYF). Subsequently, the ethyl acetate and methanol fractions were evaporated using the rotary evaporator. The dried methanolic residue was dissolved in 10 mL of deionized water, and the ethyl acetate residue in 10 mL of 100% methanol [51]. These were both used for the total phenolic contents and antioxidant capacity measurements. 

### 3.4. Determination of Total Phenolic Compounds (TPC), Flavonoid (TF) and Anthocyanin (ACY) Contents

The total phenolic compound (TPC) content of the extracts was determined using the Folin-Ciocalteu (FC) assay [52]. Briefly, 500 μL aqueous methanolic extract was mixed with 975 μL 2% Na_2_CO_3_ (*w*/*v*) and 25 μL Folin-Ciocalteu’s (FC) reagent. Deionized water (500 µL) was used as a blank. After cooling at room temperature (25 °C) for 30 min, the reaction mixture was measured against a blank at 750 nm using a UV–VIS spectrophotometer (Evolution^TM^ 201/220, Thermo Fisher Scientific, Madison, WI, USA). The results were expressed as mg gallic acid equivalent (GAE) per kg fresh weight basis (fw).

The total flavonoid (TF) content was determined using the aluminium chloride (AlCl_3_) colorimetric method described by Huang et al. [53]. Quercetin was used to prepare the standard calibration curve. Briefly, a 500 μL sample diluted with deionized water was mixed with 500 μL (2% *w*/*v*) of AlCl_3_. This mixture was kept for 30 min at room temperature, after which the absorbance of the reaction mixture was measured at 415 nm against the blank using the spectrophotometer. The TF content was expressed as mg quercetin equivalent (QE) per kg fw.

The total anthocyanin contents (ACY) of the eggplant fruit extracts were estimated spectrophotometrically according to Giusti et al. [54]. The anthocyanin content was calculated using the equation: anthocyanin content (g kg^−1^ fw) = *A* × MW × DF/(*ϵ* × *W*); where: *A* = absorbance (*A*_520nm_ − *A*_700nm_)_pH1.0_ − (*A*_520nm_ − *A*_700nm_)_pH4.5_, MW = molecular weight of delphinidin-3-*O*-glucoside (C_21_H_21_O_12_, 465.4, del-3-glc), DF = dilution factor, *ϵ* = molar absorptivity (27,000 M^−1^ cm^−1^), and *W* = sample weight (kg). The results were expressed as g del-3-glc kg^−1^ on a fresh-weight basis (fw). 

### 3.5. Determination of Antioxidant Capacity (AC)

The DPPH (2,2-diphenyl-1-picrylhydrazyl) radical scavenging activity was assayed colorimetrically using the Blois [55] method. Briefly, 100 μL of each extract was added to 1 mL DPPH solution (100 μL/mL in methanol). The mixture was then kept for 30 min in the dark, after which the absorbance was read at 520 nm using the spectrophotometer. The results were expressed as μmoL Trolox equivalent (TE) per kg fw. 

The oxygen radical absorbance capacity (ORAC) method based on a report by Ou et al. [56] was used with slight modifications. Initially, 25 μL of antioxidant (Trolox or test sample) and 100 μL of 500 nM fluorescein were placed into each well of a 96-well microplate. 2,2′-Azobis(2-methylpropionamidine)dihydrochloride (AAPH) solution (25 μL of 250 mM) was then rapidly added, and the microplate was shaken for 5 s before the first reading. The fluorescence (excitation and emission wavelengths 485 and 510 nm, respectively) was recorded every 3 min for 90 min using a Multiskan Ascent (Labystems, Helsinki, Finland) instrument. Final the ORAC values were calculated using the net area under the curve and were expressed as μmoL TE per kg fw. 

### 3.6. UHPLC-MS/MS Determination of Phenolic Acids in Eggplants

The phenolic acids of the peel and pulp of each eggplant cultivar were fractionated as free, esterified, glycosided and ester-bounded phenolic acids using previously described methods [57,58]. One gram of the pulverized and dried peel or pulp of the eggplant samples was first extracted using aqueous methanol (80:20, *v*/*v*, 3 × 20 mL) including 2,6-di-tert-butyl-p-cresol (DBC; 6 mg/100 mL). The extraction was performed in triplicate until the solution became colorless. The combined homogenates were centrifuged at 6000 rpm for 15 min at 4 °C. The supernatants were concentrated in the rotary evaporator under reduced pressure at 35 °C. The slurry was freeze-dried in the lyophilizer, and re-dissolved in water acidified to pH 2 with HCl. The free phenolic acids were extracted into diethylether. The remaining aqueous phase was split into two parts, hydrolyzed by either 2 mol/L NaOH or 6 mol/L HCl, and extracted with diethylether after adjustment to pH 2. The sediment was hydrolyzed by either 2 mol/L NaOH or 6 mol/L HCl and extracted with diethylether after adjustment to pH 2. Analytes were quantified using deuterium-labeled internal standards of 4-hydroxybenzoic (2,3,5,6-D_4_) and salicylic (3,4,5,6-D_4_) acids, as described previously [57,58], with some slight modifications. 

An ACQUITY Ultra Performance LC™ system (Waters, Milford, MA, USA) linked to a Micromass Quattro micro™ API benchtop triple quadrupole mass spectrometer (Waters MS Technologies, Manchester, UK) was used for the analysis. Samples were injected onto the BEH C8 reversed-phase column (1.7 µm, 2.1 × 150 mm, Waters, Milford, MA, USA). Phenolic acids were identified and quantified according to Gruz et al. [58]. Deuterium labeled 4-hydroxybenzoic-D4 (2,3,5,6-D_4_) and salicylic-D_4_ acids were used as internal standards in order to quantify phenolic acids. 

### 3.7. Extraction and HPLC-DAD/ESI-MS Analysis of Anthocyanins in Eggplants

The extraction of anthocyanins in the eggplant samples was performed as described by Lee et al. [59]. The lyophilized fruit material (0.2 g) was extracted in triplicate with 30 mL of 1% HCI in 40% methanol by shaking in the dark for 24 h at room temperature. The acidic methanol extract was centrifuged at 1200 rpm for 10 min at 4 °C and an aliquot of the supernatant was filtered through a Whatman No. 2 filter paper and a 0.45 µm syringe filter. 

The anthocyanin quantitative analysis was conducted following the method described by Lee et al. [59]. A HPLC was performed using an Agilent 1200 series (Waldbronn, Germany) quaternary pump, an Agilent 1200 series diode array detector, a wellplate autosampler and ChemStation software (version B.04.03). The peak area of the delphinidin-3-*O*-rutinoside standard solution was plotted against the concentration. The stock solution was prepared with a 1% TFA (*v*/*v*) in methanol to yield a 1 mg/mL concentration. Calibration curves were prepared at six different concentrations (1, 5, 10, 20, 50 and 100 μg/mL). High linearity (r^2^ > 0.999) was obtained for the standard curve. A 20 μL sample of acidic methanol extract from the eggplant was injected onto an analytical reversed-phase C18 column (TOSOH ODS 120T; 150 mm × 4.6 mm, 5μm, Tosoh Corporation, Tokyo, Japan). The mobile phase was composed of 5% formic acid in water (eluent A) and 5% formic acid in acetonitrile (eluent B). The gradient elution conditions for the HPLC-DAD were 0 min, 10% B; 20 min, 30% B; and 25 min, 60% B. The total running time was 37 min, and the flow rate was set at 0.7 mL/min. The column temperature was 30 °C, and the anthocyanins were detected by monitoring the elution at 525 nm. A HPLC-ESI-MS analysis for the identification of the anthocyanins was performed using an Agilent 1200 series HPLC system coupled with an Agilent 6110 Quadrupole mass spectrometer (Boeblingen, Germany) equipped with an electrospray ionization (ESI) source mass analyzer. Data acquisition and processing were performed on ChemStation LC and LC-MS software (version B.04.03). The mass spectrometer conditions were capillary voltage, 4000 V; fragmentation voltage, 150 V; drying gas temperature, 350 °C; gas flow of N_2_, 12 L/min; and nebulizer pressure, 50 psi. The instrument was operated in positive ion mode, scanning from *m*/*z* 100 to 1000 at a scan rate of 1.45 s/cycle. 

### 3.8. α-Amylase Inhibition Assay

An inhibition of α-amylase was performed as described by Phan et al. [60] and Esatbeyoglu et al. [61]. Approximately 10 mg of dried eggplant sample (peel and pulp) was diluted in 200 μL methanol. Next, 800 μL water was added and the mixture was vortexed, incubated for 5 min in an ultrasonic bath and centrifuged at 15,000 rpm for 10 min at 5 °C. Water was used as the blank sample, and acarbose as the standard (0.005–0.5 mg/mL). Eighty microliters of the test samples were mixed with an 80 µL 1% starch solution (prepared in 20 mM Na_3_PO_4_, 6.7 mM NaCl, pH 6.9), with an 80 µL α-amylase solution (from human saliva, Sigma Aldrich, Steinheim, Germany; 10 units/mL in water) being added only in the test group. The solutions were incubated in a ThermoMixer^®^ (Eppendorf, Germany) for 3 min at 20 °C (400 rpm). All samples were mixed with 80 µL 1% 3,5-dinitrosalicylic acid (DNS, 100 mL containing 1 g DNS and 30 g Na-K tartrate tetrahydrate dissolved in 20 mL 2 M NaOH). α-Amylase (80 µL) was added only to the control group. All samples were boiled for 15 min (the first 5 min were shaken at 400 rpm) at 99 °C and cooled in a fridge for 5 min, after which 320 µL water was added. To each well of a 96-well microplate, 160 µL of this mixture was then added and the absorbance was measured at 540 nm using a TECAN infinite M200 spectrophotometer (Tecan, Männedorf, Switzerland). Three independent experiments were performed.

### 3.9. Statistical Analysis

All analyses were performed using a completely randomized design. Three biological replicates, each with three technical replicates (*n* = 6) were performed for the peel and pulp. All data were subjected to a one-way analysis of variance (ANOVA) and the significance of the differences in contents of the compounds/chemical components was evaluated using Duncan’s Multiple Range Test with a significance threshold of *p* < 0.05. A statistical software package was also used to perform a principal component analysis (PCA) (Addinsoft 2019, XLSTAT and Data Analysis Solution, version 2019.3.2., New York, NY, USA). The correlation coefficients (*r*) were determined for the phenolic contents (total phenolic contents, phenolic acids, and anthocyanins, etc.) and antioxidant capacity values levels, comparing the mean peel and pulp values. 

## 4. Conclusions

Our results suggest the presence of significant diversity in the peel and pulp of Turkish eggplants in terms of the TPC, TF and ACY contents, antioxidant capacity (ORAC) values, phenolic acids in free, ester, glycoside and ester-bound forms, and the anthocyanin composition. The peel of black (Aydin Siyahi) and purple (Kadife Kemer) eggplants had higher phenolic contents and constituents and also higher antioxidant capacity than the white-coloured eggplant (Trabzon Kadife). This study also facilitated the identification of eggplant cultivars with high antioxidant capacity and phenolic constituents that can be recommended for consumption or used as a starting material for the improvement of eggplant antioxidant capacity by breeding.

## Figures and Tables

**Figure 1 molecules-27-02410-f001:**
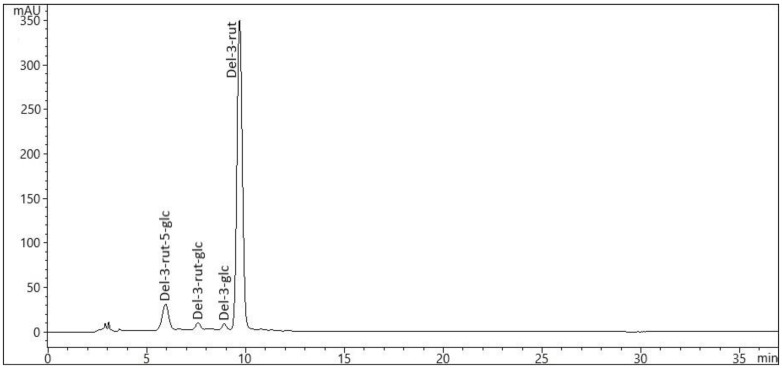
HPLC-DAD chromatograms of anthocyanins (λ = 525 nm) in the peel and pulp of eggplant fruits (‘Aydin Siyahi’, ‘Kadife Kemer’ and ‘Trabzon Kadife’). Del-3-rut-5-glc: delphinidin-3-rutinoside-5-glucoside; Del-3-rut-glc: delphinidin-3-rutinoside-glucoside; Del-3-glc: delphinidin-3-glucoside; Del-3-rut: delphinidin-3-rutinoside.

**Figure 2 molecules-27-02410-f002:**
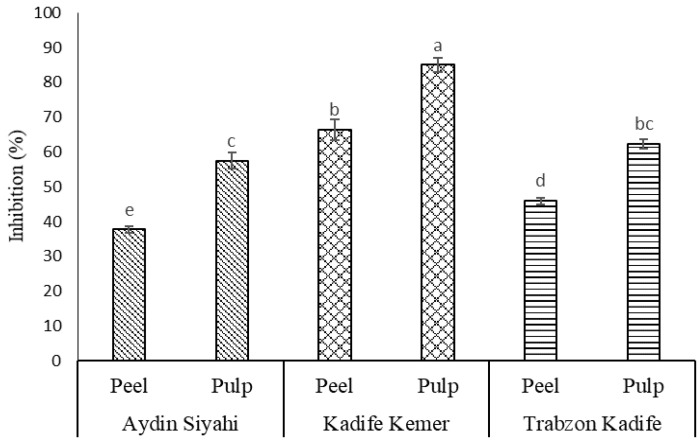
A comparison of α-amylase inhibition activities of crude phenolic extracts in fruits of common eggplants. Acarbose was used as a positive control (IC_50_ = 0.0914 ± 0.0076 mg/mL). Values are the means ± SD of three independent experiments. Values represent the mean ± SD of three independent extractions and determinations. An analysis of variance (one-way ANOVA) was used for comparisons. In each column, different small letters indicate significant differences (*p* < 0.05).

**Figure 3 molecules-27-02410-f003:**
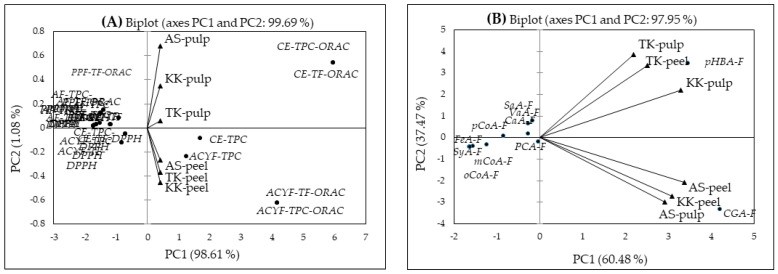
Biplot of the principal component analysis (PCA) of the peel and pulp in the eggplants; total phenolic contents and antioxidant capacity values (**A**); phenolic acids in free (**B**), ester (**C**), glycoside (**D**), and ester-bound (**E**) forms and their combined PCA model (**F**). GaA: gallic acid; PCA: protocatechuic acid; GeA: gentisic acid; *p*-HBA: *p*-hydroxybenzoic acid; VaA: vanillic acid; SaA: salicylic acid; SyA: syringic acid; CaA: caffeic acid; *p*-CoA: *p*-coumaric acid; SiA: sinapic acid; FeA: ferulic acid; *m*-CoA: *m*-coumaric acid; *o*-CoA: *o*-coumaric acid; CGA: chlorogenic acid. AS: ‘Aydin Siyahi’; KK: ‘Kadife Kemer’; TK: ‘Trabzon Kadife’. ▲: variables; ●: observations.

**Figure 4 molecules-27-02410-f004:**
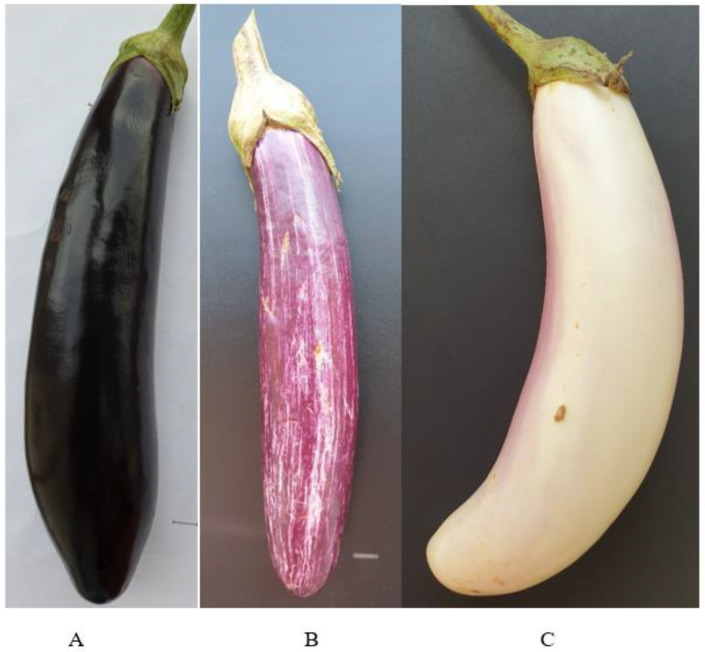
Eggplant cultivars (**A**) ‘Aydin Siyahi’, (**B**) ‘Kadife Kemer’, and (**C**) ‘Trabzon Kadife’. Scale bar = 1 cm.

**Table 1 molecules-27-02410-t001:** Comparison of total phenolic compounds content and antioxidant capacity values in the peel and pulp of common eggplant fruits *.

Parameters	‘Aydin Siyahi’ (Black)		‘Kadife Kemer’ (Purple)		‘Trabzon Kadife’ (White)	
	Peel	Pulp	Peel	Pulp	Peel	Pulp
Total Phenolic Compounds (TPC) ^‡^						
Crude Phenolic (CE)	17,193.44 ± 48.84 Fd	6551.92 ± 26.743 Cd	12,329.44 ± 52.24 Ed	6027.75 ± 42.37 Bd	8724.73 ± 27.07 Dd	3875.04 ± 3121 Ad
Aqueous Fraction (AF)	353.53 ± 0.47 Ca	650.62 ± 1.84 Fb	306.03 ± 0.60 Ba	641.05 ± 3.71 Eb	288.56 ± 7.01 Aa	514.10 ± 8.72 Db
Polyphenolic Fraction (PPF)	1341.20 ± 10.74 Fb	609.53 ± 6.02 Da	630.68 ± 12.46 Eb	348.16 ± 4.25 Ba	507.67 ± 3.47 Cb	269.04 ± 5.16 Aa
Anthocyanin Fraction (ACYF)	16,119.47 ± 49.74 Fc	5397.43 ± 28.96 Cc	11,586.29 ± 66.23 Ec	5089.54 ± 9.19 Bc	8010.09 ± 71.38 Dc	2917.69 ± 14.83 Ac
Total Flavonoids (TF) ^§^						
Crude Phenolic (CE)	3019.41 ± 20.65 Ed	1159.74 ± 18.27 Bd	1883.96 ± 56.74 Dd	1130.19 ± 41.25 Bd	1371.11 ± 22.32 Cd	632.53 ±15.44 Ad
Aqueous Fraction (AF)	22.50 ± 0.24 Fa	21.60 ± 0.15 Da	19.38 ± 0.09 Ca	20.46 ± 0.37 Ca	13.10 ± 0.13 Ba	10.11 ± 0.08 Aa
Polyphenolic Fraction (PPF)	180.88 ± 3.96 Fb	85.92 ± 2.04 Db	98.05 ± 1.30 Eb	34.23 ± 0.97 Ab	54.67 ± 0.64 Bb	65.84 ± 1.38 Cb
Anthocyanin Fraction (ACYF)	2883.76 ± 58.41 Fc	1064.60 ± 25.40 Bc	1742.59 ± 10.3 EDc	1088.13 ± 44.09 Cc	1322.73 ± 37.08 Dc	614.79 ± 5.69 Ac
DPPH ^¶^						
Crude Phenolic (CE)	8156.0 ± 81.76 Fd	2334.53 ± 35.35 Cd	3366.18 ± 103.22 Ed	1453.16 ± 21.94 Bd	2634.13 ± 17.81 Dc	662.39 ± 27.35 Ad
Aqueous Fraction (AF)	5.47 ± 0.28 Ba	8.16 ± 0.34 Ca	4.87 ± 0.22 Ba	10.16 ± 0.13 Da	4.16 ± 0.36 Aa	10.61 ± 0.61 Da
Polyphenolic Fraction (PPF)	579.51 ± 3.89 Fb	291.32 ± 9.96 Eb	265.34 ± 7.30 Db	131.25 ± 3.85 Ab	218.53 ± 6.47 Cb	203.22 ± 9.45 Bb
Anthocyanin Fraction (ACYF)	7646.63 ± 27.84 Fc	1872.22 ± 14.05 Cc	3210.73 ± 38.79 Ec	1080.95 ± 20.22 Bc	2608.37 ± 73.41 Dc	541.24 ± 8.65 Ac
ORAC ^¶^						
Crude Phenolic (CE)	37,886.96 ± 124.19 Fd	17,648.20 ± 22.19 Dd	22,670.88 ± 16.19 Ed	14,352.79 ± 131.19 Bd	19,554.97 ± 101.19 Cd	8392.16 ± 42.19 Ad
Aqueous Fraction (AF)	1217.64 ± 14.60 Da	990.43 ± 37.42 Ca	522.12 ± 28.61 Aa	899.34 ± 6.17 Ba	505.68 ± 38.59 Aa	516.39 ± 15.27 Aa
Polyphenolic Fraction (PPF)	5036.76 ± 14.80 Fb	1878.19 ± 6.86 Db	2607.07 ± 22.87 Eb	1559.02 ± 5.54 Cb	1314.20 ± 10.88 Bb	915.11 ± 12.39 Ab
Anthocyanin Fraction (ACYF)	31,929.87 ± 244.71 Ec	9718.59 ± 155.84 Bc	20,871.84 ± 330.72 Dc	9484.73 ± 180.31 Bc	17,676.07 ± 207.51 Cc	6482.34 ± 21.79 Ac
Total Anthocyanin (ACY) ^∞^	16,835.45 ± 53.42 Fe	6167.11 ± 33.71 Cc	12,081.72 ± 42.51 Ee	5715.37 ± 18.64 Bb	8486.52 ± 63.62 Dd	3218.81 ± 13.56 Aa

* Values represent the mean ± SD of three independent extractions and determinations. An analysis of variance (one-way ANOVA) was used for the comparisons. In each row, different capital letters compare statistical differences between the peel and pulp (*p* < 0.05). In each column, different small letters mean significant differences (*p* < 0.05) among the phenolic contents (TPC and TF) and antioxidant capacity (DPPH and ORAC) for each extract/subextract of the peel and pulp. ^‡^ mg gallic acid equivalent (GAE) kg^−1^ fw. ^§^ mg quercetin equivalent (QE) kg^−1^ fw. ^¶^ µmoL Trolox equivalent (TE) kg^−1^ fw. ^∞^ Total anthocyanin content (ACY) expressed as grams of delphinidin-3-*O*-glucoside (del-3-glc, MW = 500.84 g/moL and extinction coefficient = 27,000 M^−1^ cm^−1^) kg^−1^ fw.

**Table 2 molecules-27-02410-t002:** Phenolic acid contents (mg/kg fresh weight) in the fruits of common eggplants *.

	‘Aydin Siyahi’, Black		‘Kadife Kemer’, Purple		‘Trabzon Kadife’, White	
Phenolic Acids	Peel	Pulp	Peel	Pulp	Peel	Pulp
	Free (F) Form					
Protocatechuic acid ^φ^	0.96 ± 0.05 C	1.05 ± 0.03 C	3.09 ± 0.08 D	0.64 ± 0.01 B	0.60 ± 0.03 B	0.41 ± 0.02 A
*p*-Hydroxybenzoic acid ^φ^	2.15 ± 0.15 B	0.65 ± 0.03 A	5.57 ± 0.17 D	2.06 ± 0.07 B	5.00 ± 0.05 C	2.07 ± 0.03 B
Vanilic acid ^φ^	0.51 ± 0.03 B	0.18 ± 0.01 A	0.99 ± 0.09 D	0.83 ± 0.02 C	0.70 ± 0.06 C	0.81 ± 0.02 C
Salicylic acid ^φ^	0.63 ± 0.05 D	0.17 ± 0.01 A	0.50 ± 0.07 C	0.80 ± 0.01 E	0.34 ± 0.04 B	0.81 ± 0.01 E
Syringic acid ^φ^	0.02 ± 0.00 A	0.06 ± 0.00 BC	0.25 ± 0.00 D	0.05 ± 0.00 B	0.07 ± 0.00 C	0.05 ± 0.00 B
Caffeic acid ^δ^	0.25 ± 0.01 A	0.61 ± 0.03 C	3.77 ± 0.03 E	0.56 ± 0.06 C	1.05 ± 0.01 D	0.43 ± 0.03 B
*p*-Coumaric acid ^δ^	0.42 ± 0.05 BC	0.17 ± 0.00 A	0.59 ± 0.08 D	0.46 ± 0.03 C	0.34 ± 0.02 B	0.36 ± 0.01 B
Ferulic acid ^δ^	0.11 ± 0.00 A	0.29 ± 0.05 B	1.22 ± 0.13 C	0.17 ± 0.00 A	0.13 ± 0.01 A	0.19 ± 0.01 AB
*m*-Coumaric acid ^δ^	0.03 ± 0.00 D	0.02 ± 0.00 B	0.03 ± 0.00 C	0.02 ± 0.00 A	0.03 ± 0.00 C	0.02 ± 0.00 B
*o*-Coumaric acid ^δ^	0.06 ± 0.00 B	0.01 ± 0.00 A	0.11 ± 0.01 C	0.01 ± 0.00 A	0.06 ± 0.00 B	0.02 ± 0.00 A
	Ester (E) Form					
Gallic acid ^φ^	458.47 ± 5.53 E	144.81 ± 0.54 D	72.37 ± 4.40 C	8.19 ± 0.34 A	14.87 ± 0.6 B	10.40 ± 1.36 AB
Protocatechuic acid ^φ^	50.42 ± 0.10 F	35.40 ± 0.53 E	31.35 ± 1.27 D	9.93 ± 0.16 A	28.39 ± 0.25 C	27.15 ± 0.92 B
Gentisic acid ^φ^	ND	ND	ND	ND	ND	0.02 ± 0.00
*p*-Hydroxybenzoic acid ^φ^	30.37 ± 0.18 E	8.01 ± 0.39 C	10.38 ± 0.31 D	3.32 ± 0.13 A	7.04 ± 0.22 B	3.44 ± 0.26 A
Vanilic acid ^φ^	1.17 ± 0.15 A	1.50 ± 0.13 B	2.15 ± 0.16 C	1.16 ± 0.13 A	1.15 ± 0.12 A	1.19 ± 0.05 A
Syringic acid ^φ^	0.28 ± 0.00 A	0.28 ± 0.01 A	0.81 ± 0.08 C	0.97 ± 0.12 D	0.64 ± 0.02 B	0.56 ± 0.02 C
Salicylic acid ^φ^	0.57 ± 0.04 D	0.39 ± 0.03 B	0.30 ± 0.02 A	12.44 ± 0.03 E	0.33 ± 0.02 A	0.48 ± 0.00 C
Caffeic acid ^δ^	2620.35 ± 0.78 E	2512.26 ± 18.34 D	2830.44 ± 20.34 F	1250.97 ± 37.80 A	2136.59 ± 11.72 B	2457.75 ± 0.35 C
*p*-Coumaric acid ^δ^	37.39 ± 0.53 B	55.48 ± 1.61 E	41.89 ± 2.36 C	23.36 ± 1.17 A	34.60 ± 1.43 B	46.94 ± 2.17 D
Sinapic acid ^δ^	16.72 ± 0.13 A	79.59 ± 0.71 C	46.73 ± 1.74 B	91.68 ± 2.63 D	153.04 ± 2.82 F	138.97 ± 1.91 E
Ferulic acid ^δ^	243.36 ± 3.40 C	341.41 ± 3.02 E	176.08 ± 2.98 A	289.64 ± 20.71 D	212.07 ± 0.01 B	297.74 ± 1.73 D
*m*-Coumaric acid ^δ^	0.18 ± 0.00 A	0.18 ± 0.02 A	0.22 ± 0.01 B	0.16 ± 0.01 A	0.22 ± 0.00 B	0.27 ± 0.00 C
*o*-Coumaric acid ^δ^	ND	ND	ND	ND	ND	0.01 ± 0.00
	Glycoside (G) Form					
Gallic acid ^φ^	399.07 ± 19.01 B	17.49 ± 0.27 A	17.34 ± 0.18 A	7.89 ± 0.59 A	14.22 ± 0.16 A	13.09 ± 0.23 A
Protocatechuic acid ^φ^	43.60 ± 0.04 F	38.91 ± 0.65 E	24.47 ± 0.05 B	14.64 ± 0.35 A	26.88 ± 0.79 C	30.81 ± 0.06 D
Gentisic acid ^φ^	1.14 ± 0.09 B	0.63 ± 0.01 A	0.26 ± 0.01 A	9.79 ± 0.06 D	0.49 ± 0.00 A	2.86 ± 0.02 C
*p*-Hydroxybenzoic acid	85.19 ± 3.60 E	51.75 ± 0.60 D	18.57 ± 0.14 B	9.82 ± 0.25 A	16.38 ± 0.06 B	37.74 ± 0.23 C
Vanilic acid ^φ^	7.31 ± 0.29 C	6.22 ± 0.28 B	3.58 ± 0.05 A	9.12 ± 0.07 D	3.59 ± 0.11 A	9.26 ± 0.45 D
Syringic acid ^φ^	1.19 ± 0.03 B	1.99 ± 0.04 D	0.83 ± 0.02 A	3.87 ± 0.09 E	1.71 ± 0.09 C	6.68 ± 0.07 F
Salicylic acid ^φ^	0.26 ± 0.02 B	0.15 ± 0.00 AB	0.06 ± 0.00 A	9.17 ± 0.19 D	0.21 ± 0.01 AB	1.30 ± 0.01 C
Caffeic acid ^δ^	611.86 ± 8,65 E	195.00 ± 5.75 B	220.36 ± 2.45 C	9.17 ± 0.19 D	0.21 ± 0.01 AB	1.30 ± 0.01 C
*p*-Coumaric acid ^δ^	7.10 ± 0.28 D	2.95 ± 0.02 B	2.40 ± 0.07 A	8.40 ± 0.09 E	3.33 ± 0.09 C	12.97 ± 0.02 F
Sinapic acid ^δ^	1.09 ± 0.04 D	0.72 ± 0.00 B	0.81 ± 0.02 C	0.70 ± 0.00 B	0.21 ± 0.00 A	1.84 ± 0.02 E
Ferulic acid ^δ^	10.66 ± 0.20 A	31.42 ± 0.38 D	48.02 ± 0.72 E	29.83 ± 0.46 C	21.79 ± 0.48 B	59.86 ± 0.83 F
*m*-Ccoumaric acid ^δ^	0.04 ± 0.00 C	0.03 ± 0.00 B	0.01 ± 0.00 A	0.03 ± 0.00 B	0.05 ± 0.00 D	0.04 ± 0.00 C
*o*-Coumaric acid ^δ^	0.04 ± 0.00 E	0.03 ± 0.00 D	0.02 ± 0.00 C	0.01 ± 0.00 B	0.00 ± 0.00 A	0.02 ± 0.00 C
	Ester-Bound (EB) Form					
Gallic acid ^φ^	181.16 ± 1,60 E	19.47 ± 0.42 D	11.64 ± 0.34 B	1.42 ± 0.02 A	14.65 ± 0.22 C	1.34 ± 0.03 A
Protocatechuic acid ^φ^	39.09 ± 0.72 F	28.99 ± 0.38 E	21.21 ± 0.42 C	5.19 ± 0.06 A	24.45 ± 0.57 D	14.00 ± 0.11 B
*p*-Hydroxybenzoic acid ^φ^	41.51 ± 0.60 E	8.73 ± 0.38 B	12.17 ± 0.27 D	3.00 ± 0.10 A	10.12 ± 0.11 C	2.66 ± 0.06 A
Vanilic acid ^φ^	1.58 ± 0.06 C	1.52 ± 0.00 C	3.79 ± 0.01 E	1.94 ± 0.08 D	1.31 ± 0.04 B	0.82 ± 0.03 A
Syringic acid ^φ^	0.20 ± 0.00 A	0.32 ± 0.00 B	1.50 ± 0.04 E	1.10 ± 0.04 D	0.88 ± 0.03 C	0.35 ± 0.00 B
Salicylic acid ^φ^	1.23 ± 0.01 D	0.87 ± 0.02 B	1.15 ± 0.00 C	13.16 ± 0.08 E	0.88 ± 0.01 B	0.50 ± 0.07 A
Caffeic acid ^δ^	2384.72 ± 24.34 E	1824.14 ± 24.98 C	2593.88 ± 8.71 D	1072.75 ± 11.66 A	2067.31 ± 31.95 D	1122.41 ± 5.52 B
*p*-Coumaric acid ^δ^	70.30 ± 1.02 F	47.55 ± 0.13 C	52.37 ± 0.60 D	17.24 ± 0.05 A	56.05 ± 1.02 E	22.58 ± 0.22 B
Sinapic acid ^δ^	45.67 ± 1.21 B	103.91 ± 2.97 D	16.97 ± 0.39 A	104.22 ± 3.24 D	234.92 ± 9.87 E	84.51 ± 1.58 C
Ferulic acid ^δ^	335.09 ± 6.71 F	265.41 ± 1.85 D	273.39 ± 1.06 E	117.99 ± 0.51 B	197.17 ± 8.26 C	93.73 ± 1.58 A
*m*-Coumaric acid ^δ^	0.33 ± 0.00 C	0.20 ± 0.00 A	0.40 ± 0.00 D	0.20 ± 0.00 A	0.28 ± 0.00 B	0.20 ± 0.00 A

* Values represent the mean ± SD of three independent extractions and determinations. An analysis of variance (one-way ANOVA) was used for the comparisons. In each row, different capital letters compare statistical differences between the peel and pulp (*p* < 0.05) or each phenolic acid form. ^φ^ Hydroxybenzoic acid derivatives (HBAs), ^δ^ Hydroxycinnamic acid derivatives (HCAs), ND: not determined.

**Table 3 molecules-27-02410-t003:** Chlorogenic acid contents (mg/kg fresh weight) in the fruits of common eggplants *.

	‘Aydin Siyahi’, Black		‘Kadife Kemer’, Purple		‘Trabzon Kadife’, White	
Chlorogenic Acid (CGA)	Peel	Pulp	Peel	Pulp	Peel	Pulp
Free form (F)	6.08 ± 0.44 D **	4.71 ± 0.10 C	27.55 ± 0.82 E	1.14 ± 0.11 B	0.69 ± 0.03 AB	0.22 ± 0.02 A
Ester form (E)	7.82 ± 0.36 E	3.06 ± 0.23 D	0.46 ± 0.09 BC	0.00 ± 0.00 A	0.54 ± 0.07 C	0.15 ± 0.00 AB
Glycoside form (G)	294.05 ± 4.81 F	52.28 ± 0.94 B	108.16 ± 1.26 D	60.12 ± 0.20 C	213.45 ± 2.67 E	15.68 ± 0.54 A
Ester-bound form (EB)	2.41 ± 0.23 E	1.23 ± 0.04 D	1.00 ± 0.01 C	0.29 ± 0.02 B	0.43 ± 0.01 B	0.10 ± 0.00 A

* Values represent the mean ± SD of three independent extractions and determinations. ** different capital letters in each row differ significantly at *p* < 0.05 (one-way ANOVA was used for comparisons).

**Table 4 molecules-27-02410-t004:** HPLC chromatographic characteristics and delphinidin-3-*O*-rutinoside (del-3-rut) concentrations (mg/kg fw) in fruits of common eggplants using HPLC-DAD/ESI-MS ^χ^.

Anthocyanin ^φ^	Peak No	Retention Time (R_t_, min)	λ_max_ (nm)	MS [M + H]^+^	‘Aydin Siyahi’, Black		‘Kadife Kemer’, Purple		‘Trabzon Kadife’, White	
					Peel	Pulp	Peel	Pulp	Peel	Pulp
del-3-rut-5-glc	1	5.9	525	773.1	trace ^ψ^	Trace	trace	trace	trace	trace
del-3-rut-glc	2	7.6	522	773.1	trace	Trace	trace	trace	trace	trace
del-3-glc	3	8.9	528	465.1	trace	Trace	trace	trace	trace	trace
del-3-rut	4	9.7	526	611.1	1162.22 ± 5.56 D *	194.62 ± 4.54 B	336.59 ± 11.20 C	45.45 ± 1.70 A	215.11 ± 2.9 2 B	72.44 ± 1.97 A

^χ^ Values represent the mean ± SD of three independent extractions and determinations. ^φ^ del-3-rut-5-glc: delphinidin-3-rutinoside-5-glucoside; del-3-rut-glc: delphinidin-3-rutinoside-glucoside; del-3-glc: delphinidin-3-glucoside; del-3-rut: delphinidin-3-rutinoside. ^ψ^ trace: trace anthocyanin, not quantified. * different capital letters in each row differ significantly at *p* < 0.05 (one-way ANOVA was used for comparisons).

**Table 5 molecules-27-02410-t005:** Factor (F) scores of total phenolic contents, antioxidant capacity values, and phenolic acids obtained from the principal component analysis (PCA) comparing the peel and pulp in eggplants.

Total Phenolic Contents and Antioxidant Capacity ^φ^	F1	Phenolic Acid ^δ^	F1	F2	Phenolic Acid ^δ^	F1	F2	Phenolic Acid ^δ^	F1	F2	F3	Phenolic Acid ^δ^	F1	Phenolic Acid ^δ^	F1	Phenolic Acid ^δ^	F1
CE-TPC	1.696	PCA-F	−0.040	−0.189	GaA-E	−0.312	−0.305	GaA-G	0.362	−0.610	−0.118	GaA-EB	−0.775	PCA-F	−0.618	CGA-E	−0.613
AF-TPC	−1.499	*p*-HBA-F	**3.454**	**3.452**	PCA-E	−0.657	**3.594**	PCA-G	0.076	0.629	0.854	PCA-EB	−0.822	*p*-HBA-F	−0.607	GaA-G	−0.201
PPF-TPC	−1.504	VaA-F	−0.164	0.784	GeA-E	−0.787	−0.305	GeA-G	−1.078	−0.089	−0.482	*p*-HBA-EB	−0.871	VaA-F	−0.620	PCA-G	−0.465
ACYF-TPC	1.250	SaA-F	−0.278	0.651	*p*-HBA-E	−0.744	−0.305	*p*-HBA-G	0.110	0.401	1.372	VaA-EB	−0.917	SaA-F	−0.621	GeA-G	−0.603
CE-TF	−1.157	SyA-F	−1.568	−0.407	VaA-E	−0.781	−0.305	VaA-G	−0.936	0.020	−0.152	SyA-EB	−0.922	SyA-F	−0.624	*p*-HBA-G	−0.440
AF-TF	−1.728	CaA-F	−0.273	0.170	SyA-E	−0.784	−0.305	SyA-G	−1.175	−0.206	−0.240	SaA-EB	−0.904	CaA-F	−0.618	VaA-G	−0.586
PPF-TF	−1.701	*p*-CoA-F	−0.843	0.076	SaA-E	−0.768	−0.306	SaA-G	−1.113	−0.148	−0.554	CaA-EB	**8.055**	*p*-CoA-F	−0.622	SyA-G	−0.611
ACYF-TF	−1.186	FeA-F	−1.231	−0.328	CaA-E	**7.960**	0.017	CaA-G	**4.258**	**−3.325**	0.387	*p*-CoA-EB	−0.718	FeA-F	−0.622	SaA-G	−0.606
CE-TPC-DPPH	−0.704	*m*-CoA-F	−1.641	−0.443	*p*-CoA-E	−0.636	−0.299	*p*-CoA-G	−0.974	0.113	0.025	SiA-EB	−0.374	*m*-CoA-F	−0.624	CaA-G	0.338
AF-TPC-DPPH	−1.732	*o*-CoA-F	−1.631	−0.444	SiA-E	−0.438	−0.289	SiA-G	−1.306	−0.451	−0.447	FeA-EB	0.093	*o*-CoA-F	−0.624	*p*-CoA-G	−0.592
PPF-TPC-DPPH	−1.622	CGA-F	**4.215**	**−3.323**	FeA-E	0.299	−0.274	FeA-G	0.655	1.898	**2.210**	*m*-CoA-EB	−0.925	CGA-F	−0.587	SiA-G	−0.620
ACYF-TPC-DPPH	−0.821				*m*-CoA-E	−0.786	−0.305	*m*-CoA-G	−1.352	−0.513	−0.534	CGA-EB	−0.922	GaA-E	0.035	FeA-G	−0.457
CE-TPC-ORAC	**5.947**				*o*-CoA-E	−0.787	−0.305	*o*-CoA-G	−1.352	−0.515	−0.534			PCA-E	−0.200	*m*-CoA-G	−0.624
AF-TPC-ORAC	−1.394				CGA-E	−0.779	−0.305	CGA-G	**3.824**	**2.795**	−1.787			GeA-E	−0.624	*o*-CoA-G	−0.624
PPF-TPC-ORAC	−0.915													*p*-HBA-E	−0.565	CGA-G	0.141
ACYF-TPC-ORAC	**4.155**													VaA-E	−0.616	GaA-EB	−0.412
CE-TF-DPPH	−0.704													SyA-E	−0.620	PCA-EB	−0.503
AF-TF-DPPH	−1.732													SaA-E	−0.598	*p*-HBA-EB	−0.550
PPF-TF-DPPH	−1.622													CaA-E	**11.632**	VaA-EB	−0.613
ACYF-TF-DPPH	−0.821													*p*-CoA-E	−0.412	SyA-EB	−0.619
CE-TF-ORAC	**5.947**													SiA-E	−0.136	SaA-EB	−0.594
AF-TF-ORAC	−1.394													FeA-E	0.910	CaA-EB	**10.054**
PPF-TF-ORAC	−0.915													*m*-CoA-E	−0.623	*p*-CoA-EB	−0.370
ACYF-TF-ORAC	**4.155**													*o*-CoA-E	−0.624	SiA-EB	0.005
																FeA-EB	0.640
																*m*-CoA-EB	−0.623
																CGA-EB	−0.619

^φ^ For abbreviation, see Table 1. ^δ^ For abbreviation, see Table 2. F: free; E: ester; G: glycoside; EB: ester-bound.

## Data Availability

Not applicable.

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
