# Peer review of "The Phenolics and Antioxidant Properties of Black and Purple versus White Eggplant Cultivars"

_molecules, 2022, doi:10.3390/molecules27082410_

Round 1

Reviewer 1 Report

The paper is focused to the study of polyphenols and antioxidant properties of different cultivars of eggplants from Turkey. The manuscript could be valuable, but too many weaknesses arise that make not advisable its publication in the current form.

Introduction

The Introduction section should include a brief description of polyphenols classification before starting talk on anthocyanins. I addition, literature related with phenolic content of eggplants must be reviewed and updated.

-L. 62-64. This statement is inexact. See:

Molecules 2019, 24, 1536; doi:10.3390/molecules24081536 or

Metabolites 2020, 10, 209; doi:10.3390/metabo10050209

as a couple of examples.

-L.70. The expression of the results included in Table 1 is very confusing. On the one hand, the statistical treatment in the corresponding section is not explained, and, then, it is no possible to know what the different letters correspond to. On the other hand, neither treatment is not explained as a foot note, and only a column with p-values (all >0.05, with only one exception) is included. It seems not logic to study differences within a row, as results from peel and whole fruit are all together (and should be separately compared). Finally, correlation index must not be included in the same table than results. In addition, superscripts are excessively small and very difficult to see.

-L. 72-73. This is not correct according with p values at the right side of the Table.

-L. 73-75. Confusing sentence, please rewrite.

-L. 86-89. The results given in the text are not coincident with those of the Table.

-L. 142 and following and Table 2. The same consideration of Table 1 may be applied to table 2. The explanation of statistical treatment as a foot note is needed, as well as the meaning of PC1 and PC1. In the same line, the factor analysis should be separated from results.

-L. 149-151. According with p-values of the Table, just a few variables differ significantly.

-L. 217-219.Please review this information as peak 4 (del-3-rut) seems to be much higher in peel of  ‘Kadife Kemer’

-L. 230-233 and Figure 3. Where is the statistical treatment of the amylase activity between samples?

-L. 236-238. Possible reasons of the discrepancy between α-amylase inhibition activity and antioxidant properties of the samples should be exposed and discussed.

-L. 267-270. Results of the PCA are not surprising, since peel and fruits are very different in all the variables measured. Moreover, nothing is said about how this analysis has been made.

In addition of the results concerning PCA, the meaning of this analysis must be discussed. I mean, which are the reasons of doing such analysis? What essential information does it provide? What are the main conclusions?

-L. It seems that a total of 24 eggplants were used for each cultivar. Were the plants pooled before analysis? Please explain. 

-L. 355. In my opinion, the term “whole fruit” in this context is incorrect, since peel has been previously removed (according with the description). Therefore, it would be more appropriate to call it "pulp", to avoid confusions.

-L. 357. Include the negative sign before number 80.

-L- 363. One extraction followed by triple extraction make a total of four extractions… were there 3 or 4 extractions in total?

-L. 364. Replace letter x by symbol × (also L. 377 and others). There is no need of indicate speed in g and in rpm. Use rpm only when characteristics of the motor and centrifuge are given. The same in other parts of the text.

-L. 369. Check subsection number. Which aqueous extract? samples are supposed to be lyophilized. Please explain.

-L. 377. What means “3 x 3 mL deionized water”?

-L. 387-401. In this paragraph the determination of anthocyanin content is lacking.

-L. 441 and 478. Check degree symbol.

-L. 437-470. Sections 3.7 and 3.8 may be joined into just one.

-L. 328…. 489.In the Material and Methods section, the statistical analysis is lacking.

Author Response

Comments of Reviewer 1

Introduction

Reviewer comment: The Introduction section should include a brief description of polyphenols classification before starting talk on anthocyanins. I addition, literature related with phenolic content of eggplants must be reviewed and updated.

Respond to Reviewer 1: about sentence of polyphenols and eggplant information, the below paragraphs are included.

Ranking among the top 10 in terms of oxygen radical absorbing capacity (ORAC) due to the fruits’ polyphenols constituents, the cultivated eggplant (Solanum melongena L, Fam: Solanaceae) also known as aubergine, brinjal, melanzane, berenjena or patlıcan is an economically important vegetable crop of tropical and subtropical zones of the world including Asia (47 142 210 tons), Africa (1 814 535 tons), Europe (936 642 tons), the Americas (295 387 tons) and Oceania (4342 tons) [3-7]. Eggplant is grown over 1.7 million hectares world-wide (Turkey ranks the fourth in the world in eggplant production with an annual production of approximately 816 000 tons by meeting production of 2% in world production [8-9].

Eggplant exhibits wide diversity in color (purple, purple/black, green or variegated), size (ave. 210.4 g, range 42 - 464) and shape (elongated, ovoid or slender). Based on the fruit shape, eggplant is classified as egg-shaped (S. melongena var. esculentum) long and slender in shape (S. melongena var. serpentinum), or dwarf types (S. melongena var. depressum). The oblongo to elongate-shaped dark purple, purple/black or violet eggplants are is used world wide, but other varieties that differ in color, size and shape are also known [3, 4, 6, 7, 10-12].

Fruit and vegetables are food sources of a variety of non-nutritive bioactive compounds (mainly phenolics) that their long-term or regular consumption of which is associated with a reduced risk of certain types of cancers, and cardiovascular and other neurodegenerative diseases [1, 13, 14].  Polyphenols are widely distributed in nature and currently, approximately 8000 different structures of plant these compounds having been identified to date. The most common classification of phenolic metabolites based on two aromatic rings connected by a bridge consisting of three carbons (C6-C3-C6) distinguishes the flavonoid and non-flavonoid compounds. Flavonoids consist of more than 6,000 types are divided into different six main subclasses (anthocyanins, chalcones, flavanones flavones, flavonols and isoflavones), depending on the carbon of the C ring on which the B ring is attached and the degree of unsaturation and oxidation of the C ring. In the physiological state flavonoids usually occur in association with sugar as glycosides. The second class of plant phenolics—non-flavonoid metabolites—consists of the following subgroups: phenolic acids (hydroxybenzoates C6-C1, hydroxycinnamates C6-C3), lignans (C6-C3)2 and stilbenes C6-C2-C6. Two other subclasses of non-flavonoids compounds with high molecular weights that lack a defined primary carbon base and occur mainly as complicated biopolymers, being unique for the particular polyphenols are tannins (hydrolysable and condensed) and lignins [14-16].

The major phenolic compounds in eggplant fruits are reported to be highly beneficial for human health due to their known biological activities, and they can be potentially used in treatments of several metabolic and cardiovascular diseases [17]. Delphinidin derivatives are the only anthocyanins identified in eggplant fruits. The most common anthocyanin structure in peel in eggplant fruits is delphinidin-3-(p-coumaroyl-rutinoside)-5-glucoside, known as nasunin [18], while the main phenolic acid in flesh is chlorogenic acid (CGA) together with its hydroxycinnamic acid conjugates (chlorogenic acid isomers, isochlorogenic acid isomers, amide conjugates, unidentified caffeic acids conjugates, and acetylated chlorogenic acid isomers) varying from 75 to 94% of total phenolic content in a wide range of eggplant cultivars [3, 4, 6, 8-10, 18, 19].

Studies have indicated that anthocyanins and phenolic acids contribute to high antioxidant properties in eggplant [17-18]. Eggplant exhibits potential health benefits in a number of degenerative diseases, cancer, cardiovascular diseases, diabetes, pulmonary disorders and Alzheimer’s disease [6, 12, 20-23]. In addition, studies have shown that eggplant varieties enriched in phenolic phytochemicals and moderate free radical scavenging-linked antioxidant activity have a potential to reduce hyperglycemia-induced vascular complications resulting from oxidative damage [20, 24-26]. In this respect, a previous invitro study showed that several eggplant samples (e.g., White pulp and Graffiti skin) had high a-glucosidase inhibitory activity combined with low a-amylase inhibitory activity indicating a potential ability to reduce glucose absorption in the intestine [20].

Reviewer comment: -L. 62-64. This statement is inexact. See:

Molecules 2019, 24, 1536; doi:10.3390/molecules24081536 or

Plant Material and Extract Preparation

Eggplant, red onion, black carrot, purple sweet potato, and red chicory were purchased from a local market in Cluj-Napoca, Romania. The vegetables were stored at −18 â—¦C until further analysis. Acidified methanol (MeOH + 0.03% HCl) was prepared before proceeding the extraction, and the samples were fine grounded in a mortar. Subsequently, 10 grams of the grounded sample were mixed with the solvent. The colored mixture was centrifuged at 3214 g, the supernatant was collected, and the extraction procedure was repeated until the samples turned colorless. Further, the collected extracts were evaporated at 40 â—¦C under reduced pressure (Rotavapor R-124, Buchi, Switzerland), dissolved in a known amount of acidified water, and filtered through 0.45 µm Millipore filter before all of the qualitative and quantitative analyses. The extractions were carried out at room temperature.

Metabolites 2020, 10, 209; doi:10.3390/metabo10050209

Fruits from tomato (S. lycopersicum cv. “M82”) (TGRC, Tomato Genetics Resource Center, Davis, CA, USA), eggplant cv. “Ryoma” (S. melongena) (Takii, Japan), and different pepper (C. annuum) cultivars (green paprika cv. “New Ace” (Takii, Japan), red paprika cv. “Flupy Red EX” (e-taneya, Japan), purple pepper cv. “Nara Murasaki” (Tsurushin, Japan), green pepper cv. “Miogi” (e-taneya, Japan), yellow paprika cv. “Sonia Gold” (Sakata-no-Tane, Japan), and jalapeño pepper cv. “Jalapeño” (Marche, Japan)) were grown from May–December 2019 in standard soil under open field (longitude, 34.734433; latitude 135.736754) conditions. Three developmental stages were defined for analysis from fruit peel and flesh. Immature fruit, mid-stage, and final mature stages were selected and defined based on the height of each fruit and the color of peel and seeds. Immature fruit are half the height of mature fruit and green in color. In the case of purple pepper and eggplant, the immature fruit color was purple. Green mature fruit had approximately the same height as the final mature stage but still green in color. Purple pepper was also purple during its green mature stage. The final mature stage is when peel color has completely changed from green to its ripe fruit color (red, yellow). Mature eggplant fruit were distinguished through their dark brown seed color. Three independent biological replicates per plant tissue during each ripening stage were harvested and used for metabolomic analysis. After tissue separation, fruit peel and p-L.70. The expression of the results included in Table 1 is very confusing. On the one hand, the statistical treatment in the corresponding section is not explained, and, then, it is no possible to know what the different letters correspond to. On the other hand, neither treatment is not explained as a foot note, and only a column with p-values (all >0.05, with only one exception) is included. It seems not logic to study differences within a row, as results from peel and whole fruit are all together (and should be separately compared). Finally, correlation index must not be included in the same table than results. In addition, superscripts are excessively small and very difficult to see.

Respond to Reviewer 1: In the articles, the origin of the eggplants (cultivar, variety, genotype, etc.) is unknown. In my opinion, the authors Frond et al. (209) would have used a common black eggplant-‘Aydin Siyahi’, but for a scientific point of view, the ciation of the article with an unknown eggplant sources is not acceptable. In the second artcile, Japanese eggplant sources have been investigated. Therefore, as authors, we cannot accept their citation in the present manuscipt.

Reviewer comment -L.70. The expression of the results included in Table 1 is very confusing. On the one hand, the statistical treatment in the corresponding section is not explained, and, then, it is no possible to know what the different letters correspond to. On the other hand, neither treatment is not explained as a foot note, and only a column with p-values (all >0.05, with only one exception) is included. It seems not logic to study differences within a row, as results from peel and whole fruit are all together (and should be separately compared). Finally, correlation index must not be included in the same table than results. In addition, superscripts are excessively small and very difficult to see.

-L. 72-73. This is not correct according with p values at the right side of the Table.

-L. 73-75. Confusing sentence, please rewrite.

-L. 86-89. The results given in the text are not coincident with those of the Table.

-L. 142 and following and Table 2. The same consideration of Table 1 may be applied to table 2. The explanation of statistical treatment as a foot note is needed, as well as the meaning of PC1 and PC1. In the same line, the factor analysis should be separated from results.

-L. 149-151. According with p-values of the Table, just a few variables differ significantly.

Respond to Reviewer 1: -L 72-73, 73-75, 86-89,142, 149-151. Table 1 was revised, all confusing points were erased, the statistical treatment was included and explained in an understandable manner, what to different letters correspond to is given, all foot notes were included, statistical analysis-One-way ANOVA was employed to the row and the column data, correlation analysis (Pearson) was used, but the tables are not included in the text, only the r significance values/levels has been extracted and put in result setion during comparing data. Superscripts, now, are capital and easily are seen. The confusing sentences were corrected. All these changes and recommendations were applied to Table 2. The factor analysis data extracted form PCA (PC1 and PC2) were given in Table 5, more clear and more understandable manner. In case of request, the correlation matrix tables    (6 tables) can be send.

Table 1. Comparison of total phenolic compounds content and antioxidant capacity values in peel and pulp in the eggplant fuits*

Parameters

‘Aydin Siyahi’ (black)

‘Kadife Kemer’ (purple)

‘Trabzon Kadife’ (white)

Peel

Pulp

Peel

Pulp

Peel

Pulp

        Total phenolic compounds (TPC)

Crude phenolic (CE)

17193.44 ± 48.84 Fd

6551.92 ± 26.743 Cd

12329.44 ± 52.24 Ed

6027.75 ± 42.37 Bd

8724.73 ± 27.07 Dd

3875.04 ± 3121 Ad

Aqueous Fraction (AF)

353.53 ± 0.47 Ca

650.62 ± 1.84 Fb

306.03 ± 0.60 Ba

641.05 ± 3.71 Eb

288.56 ± 7.01 Aa

514.10 ± 8.72 Db

Polyphenolic Fraction (PPF)

1341.20 ± 10.74 Fb

609.53 ± 6.02 Da

630.68 ± 12.46 Eb

348.16 ± 4.25 Ba

507.67 ± 3.47 Cb

269.04 ± 5.16 Aa

Anthocyanin Fraction (ACYF)

16119.47 ± 49.74 Fc

5397.43 ± 28.96 Cc

11586.29 ± 66.23 Ec

5089.54 ± 9.19 Bc

8010.09 ± 71.38 Dc

2917.69 ± 14.83 Ac

        Total flavonoids (TF)§

Crude phenolic (CE)             

3019.41 ±20.65 Ed

1159.74 ±18.27 Bd

1883.96 ±56.74 Dd

1130.19 ±41.25 Bd

1371.11 ± 22.32 Cd

632.53 ±15.44 Ad

Aqueous Fraction (AF)

22.50 ± 0.24 Fa

21.60 ± 0.15 Da

19.38 ± 0.09 Ca

20.46 ± 0.37 Ca

13.10 ± 0.13 Ba

10.11 ± 0.08 Aa

Polyphenolic Fraction (PPF)

180.88 ± 3.96 Fb

85.92 ± 2.04 Db

98.05 ± 1.30 Eb

34.23 ± 0.97 Ab

54.67 ± 0.64 Bb

65.84 ± 1.38 Cb

Anthocyanin Fraction (ACYF)

2883.76 ± 58.41 Fc

1064.60 ± 25.40 Bc

1742.59 ± 10.3 EDc

1088.13 ± 44.09 Cc

1322.73 ± 37.08 Dc

614.79 ± 5.69 Ac

        DPPH

Crude phenolic (CE)

8156.0 ± 81.76 Fd

2334.53 ± 35.35 Cd

3366.18 ± 103.22 Ed

1453.16 ± 21.94 Bd

2634.13 ± 17.81 Dc

662.39 ± 27.35 Ad

Aqueous Fraction (AF)

5.47 ± 0.28 Ba

8.16 ± 0.34 Ca

4.87 ± 0.22 Ba

10.16 ± 0.13 Da

4.16 ± 0.36 Aa

10.61 ± 0.61 Da

Polyphenolic Fraction (PPF)

579.51 ± 3.89 Fb

291.32 ± 9.96 Eb

265.34 ± 7.30 Db

131.25 ± 3.85 Ab

218.53 ± 6.47 Cb

203.22 ± 9.45 Bb

Anthocyanin Fraction (ACYF)

7646.63 ± 27.84 Fc

1872.22 ± 14.05 Cc

3210.73 ± 38.79 Ec

1080.95 ± 20.22 Bc

2608.37 ± 73.41 Dc

541.24 ± 8.65 Ac

        ORAC

Crude phenolic (CE)

37886.96 ± 124.19Fd

17648.20 ± 22.19 Dd

22670.88 ± 16.19 Ed

14352.79 ± 131.19 Bd

19554.97 ± 101.19 Cd

8392.16 ± 42.19 Ad

Aqueous Fraction (AF)

1217.64 ± 14.60 Da

990.43 ± 37.42 Ca

522.12 ± 28.61 Aa

899.34 ± 6.17 Ba

505.68 ± 38.59 Aa

516.39 ± 15.27 Aa

Polyphenolic Fraction (PPF)

5036.76 ± 14.80 Fb

1878.19 ± 6.86 Db

2607.07 ± 22.87 Eb

1559.02 ± 5.54 Cb

1314.20 ± 10.88 Bb

915.11 ± 12.39 Ab

Anthocyanin Fraction (ACYF)

31929.87 ± 244.71 Ec

9718.59 ± 155.84 Bc

20871.84 ± 330.72 Dc

9484.73 ± 180.31 Bc

17676.07 ± 207.51 Cc

6482.34 ± 21.79 Ac

Total Anthocyanin (ACY)

16835.45 ± 53.42 Fe

6167.11 ± 33.71 Cc

12081.72 ± 42.51 Ee

5715.37 ± 18.64 Bb

8486.52 ± 63.62 Dd

3218.81 ± 13.56 Aa

*Values represent mean ± SD of three independent extractions and determinations. Analysis of variance (one-way ANOVA) was used for comparisons. In each row, different capital letters compare statistical differences between peel and pulp (P < 0.05). In each column, different small letters mean significant differences (P < 0.05) among phenolic contents (TPC and TF) and antioxidant capacity (DPPH and ORAC) for each extract/sub-extract of the peel and pulp.

mg gallic acid equivalent (GAE) kg -1 fw

  • mg quercetin equivalent (QE) kg -1 fw

µmol Trolox equivalent (TE) kg-1 fw

Total monomeric anthocyanin content (ACY) expressed as grams of delphinidin-3-O-glucoside (del-3-glc, MW = 500.84 g/mol and extinction coefficient = 27 000 M-1 cm-1) kg-1 fw

Table 2. Phenolic acid contents (mg/kg fresh weight) in fruits in common eggplants*.

‘Aydin Siyahi’, black

‘Kadife Kemer’, purple

‘Trabzon Kadife’, white

Phenolic acids

Peel

Pulp

Peel

Pulp

Peel

Pulp

Free (F) Form

Protocatechuic acidf

0.96 ± 0.05 C

1.05 ± 0.03 C

3.09 ± 0.08 D

0.64 ± 0.01 B

0.60 ± 0.03 B

0.41 ± 0.02 A

p-Hydroxybenzoic acidf

2.15 ± 0.15 B

0.65 ± 0.03 A

5.57 ± 0.17 D

2.06 ± 0.07 B

5.00 ± 0.05 C

2.07 ± 0.03 B

Vanilic acidf

0.51 ± 0.03 B

0.18 ± 0.01 A

0.99 ± 0.09 D

0.83 ± 0.02 C

0.70 ± 0.06 C

0.81 ± 0.02 C

Salicylic acidf

0.63 ± 0.05 D

0.17 ± 0.01 A

0.50 ± 0.07 C

0.80 ± 0.01 E

0.34 ± 0.04 B

0.81 ± 0.01 E

Syringic acidf

0.02 ± 0.00 A

0.06 ± 0.00 BC

0.25 ± 0.00 D

0.05 ± 0.00 B

0.07 ± 0.00 C

0.05 ± 0.00 B

Caffeic acidd

0.25 ± 0.01 A

0.61 ± 0.03 C

3.77 ± 0.03 E

0.56 ± 0.06 C

1.05 ± 0.01 D

0.43 ± 0.03 B

p-Coumaric acidd

0.42 ± 0.05 BC

0.17 ± 0.00 A

0.59 ± 0.08 D

0.46 ± 0.03 C

0.34 ± 0.02 B

0.36 ± 0.01 B

Ferulic acidd

0.11 ± 0.00 A

0.29 ± 0.05 B

1.22 ± 0.13 C

0.17 ± 0.00 A

0.13 ± 0.01 A

0.19 ± 0.01 AB

m- Coumaric acidd

0.03 ± 0.00 D

0.02 ± 0.00 B

0.03 ± 0.00 C

0.02 ± 0.00 A

0.03 ± 0.00 C

0.02 ± 0.00 B

o- Coumaric acidd

0.06 ± 0.00 B

0.01 ± 0.00 A

0.11 ± 0.01 C

0.01 ± 0.00 A

0.06 ± 0.00 B

0.02 ± 0.00 A

Ester (E) Form

Gallic acidf

458.47 ± 5.53 E

144.81 ± 0.54 D

72.37 ± 4.40 C

8.19 ± 0.34 A

14.87 ± 0.6 B

10.40 ± 1.36 AB

Protocatechuic acidf

50.42 ± 0.10 F

35.40 ± 0.53 E

31.35 ± 1.27 D

9.93 ± 0.16 A

28.39 ± 0.25 C

27.15 ± 0.92 B

Gentisic Acidf

ND

ND

ND

ND

ND

0.02 ± 0.00

p-Hydroxybenzoic acidf

30.37 ± 0.18 E

8.01 ± 0.39 C

10.38 ± 0.31 D

3.32 ± 0.13 A

7.04 ± 0.22 B

3.44 ± 0.26 A

Vanilic acidf

1.17 ± 0.15 A

1.50 ± 0.13 B

2.15 ± 0.16 C

1.16 ± 0.13 A

1.15 ± 0.12 A

1.19 ± 0.05 A

Syringic acidf

0.28 ± 0.00 A

0.28 ± 0.01 A

0.81 ± 0.08 C

0.97 ± 0.12 D

0.64 ± 0.02 B

0.56 ± 0.02 C

Salicylic acidf

0.57 ± 0.04 D

0.39 ± 0.03 B

0.30 ± 0.02 A

12.44 ± 0.03 E

0.33 ± 0.02 A

0.48 ± 0.00 C

Caffeic acidd

2620.35 ± 0.78 E

2512.26 ± 18.34 D

2830.44 ± 20.34 F

1250.97 ± 37.80 A

2136.59 ± 11.72 B

2457.75 ± 0.35 C

p-Coumaric acidd

37.39 ± 0.53 B

55.48 ± 1.61 E

41.89 ± 2.36 C

23.36 ± 1.17 A

34.60 ± 1.43 B

46.94 ± 2.17 D

Sinapic acidd

16.72 ± 0.13 A

79.59 ± 0.71 C

46.73 ± 1.74 B

91.68 ± 2.63 D

153.04 ± 2.82 F

138.97 ± 1.91 E

Ferulic acidd

243.36 ± 3.40 C

341.41 ± 3.02 E

176.08 ± 2.98 A

289.64 ± 20.71 D

212.07 ± 0.01 B

297.74 ± 1.73 D

m- Coumaric acidd

0.18 ± 0.00 A

0.18 ± 0.02 A

0.22 ± 0.01 B

0.16 ± 0.01 A

0.22 ± 0.00 B

0.27 ± 0.00 C

o- Coumaric acidd

ND

ND

ND

ND

ND

0.01 ± 0.00

Glycoside (G) Form

Gallic acidf

399.07 ± 19.01 B

17.49 ± 0.27 A

17.34 ± 0.18 A

7.89 ± 0.59 A

14.22 ± 0.16 A

13.09 ± 0.23 A

Protocatechuic acidf

43.60 ± 0.04 F

38.91 ± 0.65 E

24.47 ± 0.05 B

14.64 ± 0.35 A

26.88 ± 0.79 C

30.81 ± 0.06 D

Gentisic Acidf

1.14 ± 0.09 B

0.63 ± 0.01 A

0.26 ± 0.01 A

9.79 ± 0.06 D

0.49 ± 0.00 A

2.86 ± 0.02 C

p-Hydroxybenzoic acid

85.19 ± 3.60 E

51.75 ± 0.60 D

18.57 ± 0.14 B

9.82 ± 0.25 A

16.38 ± 0.06 B

37.74 ± 0.23 C

Vanilic acidf

7.31 ± 0.29 C

6.22 ± 0.28 B

3.58 ± 0.05 A

9.12 ± 0.07 D

3.59 ± 0.11 A

9.26 ± 0.45 D

Syringic acidf

1.19 ± 0.03 B

1.99 ± 0.04 D

0.83 ± 0.02 A

3.87 ± 0.09 E

1.71 ± 0.09 C

6.68 ± 0.07 F

Salicylic acidf

0.26 ± 0.02 B

0.15 ± 0.00 AB

0.06 ± 0.00 A

9.17 ± 0.19 D

0.21 ± 0.01 AB

1.30 ± 0.01 C

Caffeic acidd

611.86 ± 8,65 E

195.00 ± 5.75 B

220.36 ± 2.45 C

9.17 ± 0.19 D

0.21 ± 0.01 AB

1.30 ± 0.01 C

p-Coumaric acidd

7.10 ± 0.28 D

2.95 ± 0.02 B

2.40 ± 0.07 A

8.40 ± 0.09 E

3.33 ± 0.09 C

12.97 ± 0.02 F

Sinapic acidd

1.09 ± 0.04 D

0.72 ± 0.00 B

0.81 ± 0.02 C

0.70 ± 0.00 B

0.21 ± 0.00 A

1.84 ± 0.02 E

Ferulic acidd

10.66 ± 0.20 A

31.42 ± 0.38 D

48.02 ± 0.72 E

29.83 ± 0.46 C

21.79 ± 0.48 B

59.86 ± 0.83 F

m-Ccoumaric acidd

0.04 ± 0.00 C

0.03 ± 0.00 B

0.01 ± 0.00 A

0.03 ± 0.00 B

0.05 ± 0.00 D

0.04 ± 0.00 C

o-Coumaric acidd

0.04 ± 0.00 E

0.03 ± 0.00 D

0.02 ± 0.00 C

0.01 ± 0.00 B

0.00 ± 0.00 A

0.02 ± 0.00 C

Ester-Bound (EB) Form

Gallic acidf

181.16 ± 1,60 E

19.47 ± 0.42 D

11.64 ± 0.34 B

1.42 ± 0.02 A

14.65 ± 0.22 C

1.34 ± 0.03 A

Protocatechuic acidf

39.09 ± 0.72 F

28.99 ± 0.38 E

21.21 ± 0.42 C

5.19 ± 0.06 A

24.45 ± 0.57 D

14.00 ± 0.11 B

p-Hydroxybenzoic acidf

41.51 ± 0.60 E

8.73 ± 0.38 B

12.17 ± 0.27 D

3.00 ± 0.10 A

10.12 ± 0.11 C

2.66 ± 0.06 A

Vanilic acidf

1.58 ± 0.06 C

1.52 ± 0.00 C

3.79 ± 0.01 E

1.94 ± 0.08 D

1.31 ± 0.04 B

0.82 ± 0.03 A

Syringic acidf

0.20 ± 0.00 A

0.32 ± 0.00 B

1.50 ± 0.04 E

1.10 ± 0.04 D

0.88 ± 0.03 C

0.35 ± 0.00 B

Salicylic acidf

1.23 ± 0.01 D

0.87 ± 0.02 B

1.15 ± 0.00 C

13.16 ± 0.08 E

0.88 ± 0.01 B

0.50 ± 0.07 A

Caffeic acidd

2384.72 ± 24.34 E

1824.14 ± 24.98 C

2593.88 ± 8.71 D

1072.75 ± 11.66 A

2067.31 ± 31.95 D

1122.41 ± 5.52 B

p-Coumaric acidd

70.30 ± 1.02 F

47.55 ± 0.13 C

52.37 ± 0.60 D

17.24 ± 0.05 A

56.05 ± 1.02 E

22.58 ± 0.22 B

Sinapic acidd

45.67 ± 1.21 B

103.91 ± 2.97 D

16.97 ± 0.39 A

104.22 ± 3.24 D

234.92 ± 9.87 E

84.51 ± 1.58 C

Ferulic acidd

335.09 ± 6.71 F

265.41 ± 1.85 D

273.39 ± 1.06 E

117.99 ± 0.51 B

197.17 ± 8.26 C

93.73 ± 1.58 A

m-Coumaric acidd

0.33 ± 0.00 C

0.20 ± 0.00 A

0.40 ± 0.00 D

0.20 ± 0.00 A

0.28 ± 0.00 B

0.20 ± 0.00 A

*Values represent mean ± SD of three independent extractions and determinations. Analysis of variance (one-way ANOVA) was used for comparisons. In each row, different capital letters compare statistical differences between peel and pulp (P < 0.05) or each phenolic acid form. fHydroxybenzoic acid derivatives (HBAs), dHydroxycinnamic acid derivatives (HCAs),  ND.; not determined.

Reviewer comment -L. 217-219.Please review this information as peak 4 (del-3-rut) seems to be much higher in peel of  ‘Kadife Kemer’

Respond to Reviewer 1: The comment was explained in light of cultivar specific and anthocyanin biosynthesis point of view.

Figure 1 shows the presence of detected anthocyanins in the eggplants. The major anthocyanin in eggplant in the present study was delphinidin-3-rutinoside (del-3-rut) (Table 4). The LC-MS data in the present study also confirmed the presence of del-3-rut-5-glc, del-3-rut-glc and del-3-glc, although there were also trace anthocyanins. The eggplants exhibited significantly higher del-3-rut concentrations in the peel of ‘Aydin Siyahi’ (avg. 1162.22 mg/g fw), followed by ‘Kadife Kemer’ (avg. 336.59 mg/kg fw) and ‘Trabzon Kadife’ (avg. 215.11 mg/kg fw), in comparison to the pulp (range; 45.45 - 194.62 mg/kg fw). Nasunin, a delphinidin derivative (delphinidin-3-(p-coumaroylrutinoside)-5-glucoside) first reported in Japanese eggplants, is the major anthocyanin found in eggplant peel [3, 23].  These anthocyanins have also been reported as the major anthocyanins in different variety of eggplants reported from Bulgaria [33], Japan [34] and the United States [35]. Sadilova et al. [10] detected the same pattern of delphinidins (delphinidin-3-rutinoside, delphinidin-3-rutinoside-5-glucoside, delphinidin-3-glucoside) in eggplants to that earlier determined by Wu and Prior [35] in eggplants reported from the USA. In a recent study, Calumpang et al. [36] have also noticed the presence of del-3-rut (described as anthocyanin_I, delphinidin-3-O-(-feruloyl) rutinoside, in the article) in a purple Japanese (Takii) eggplant ‘Ryoma’ (S. melongena). Acylated anthocyanins (p-coumaroyl, feruloyl or caffeoyl acyl moieties) are the most abundant forms in eggplant, although in the latter some accessions are in the latter found in which a non-acylated anthocyanin, namely del-3-rut, is predominant [10, 34, 37]. However, except for del-3-rut, non-acylated anthocyanins account for only a small proportion of the total anthocyanin content. Despite the general structural similarity of anthocyanins in eggplant, deviations can be sometimes observed [18]. For instance, in peel of ‘Zi Chang’ eggplant, only two anthocyanins, delphinidin-3-glucoside-5-(coumaryl)-dirhamnoside and delphinidin-3-glucoside-5-dirhamnoside, are found in position 3 that carries a single glucose moiety instead of the common p-coumaroyl-rutinoside, while position 5 is conjugated with a dirhamnosyl moiety [38]. This suggests the existence of genetic variation for enzymes such as glycosyltransferases, which mediate the conjugation of anthocyanidins with sugar moieties [18, 38]. It Anthocyanins are reported to be involved in pigmentation, specifically purple to black pigmentation in the peel of eggplant fruit [34, 36]. The present study we compared anthocyanins in the peel of black, purple and white eggplants. Discoloration and color-changing phenomena have been observed in plant tissues during development. Anthocyanin discoloration may be due to either anthocyanin reduction in plant tissues or to structural changes in the anthocyanin that leads to a loss of color that controlled by active enzyme-driven breakdown processes [e.g. polyphenol oxidase (PPO), peroxidase (POD) and b-glucosidases] or non-enzymatic factors-attributed to either reduced biosynthesis or increased degradation of anthocyanins, or a combination of both. In anthocyanin biosynthetic pathway, expression of late biosynthetic genes [(LBGs—F3′H, F3′5′H, DFR, ANS, and UFGT)] are required for the biosynthesis of specific classes of flavonoids, including anthocyanins and determines the quantitative variation in anthocyanins. Positive correlations between expression levels of LBGs and anthocyanin content have been consistently observed in many Solanaceous vegetables, including eggplant [18]. Transcript levels of late biosynthetic genes decrease during later stages of ripening when discoloration occurs. Anthocyanin biosynthesis is regulated by MBW complexes consisting of different MYBs, but with the same bHLH and WD40 transcription factors. Reduced biosynthesis is controlled by downregulation of MYB activators and upregulation of MYB repressors. The expression level of SmCHS in eggplant has been reported to be significantly upregulated in black (Black Beauty) or violet (Classic) fruits compared to the green (genotype E13GB42) or white (Ghostbuster) mutants [39, 40]. In addition, the transcript levels of SmCHS and SmCHI, but not SmF3H, have been shown to correlate well with the anthocyanin accumulation pattern in eggplant ‘Lanshan Hexian’ [41]. Studies have also emphasized that non-enzymatic factors also have a considerable effect on the chemical structure of anthocyanins that determine anthocyanin color and stability and may enhance the vulnerability of enzymes that degrade anthocyanins. The higher the level of B-ring hydroxylation, the more purple the color, but the more unstable the anthocyanins are [42]. The effect of glycosylation varies depending on the number and the position of sugar moieties [43]. In addition, glycosylation at C3 elevates stability and shifts color slightly toward red. The stabilizing effect of diglycosides at C3 is stronger than that of monoglycosides. In contrast, glycosylation at C5 reduces pigment intensity. Acylation increases anthocyanin stability and an increasing number of acyl moieties causes a color shift from red to blue [43, 44].

Reviewer comment: -L. 230-233 and Figure 3. Where is the statistical treatment of the amylase activity between samples?

Respond to Reviewer 1: Statistical analysis was done,

Figure 2. A comparison of α-amylase inhibition activities of crude phenolics extracts (CE) in eggplants. Acarbose was used as a positive control (IC50 = 0.0914 mg/ml ± 0.0076 mg/ml). Values are means ± SD of three independent experiments. Values represent mean ± SD of three independent extractions and determinations. Analysis of variance (one-way ANOVA) was used for comparisons. In each column, different small letters indicate significant differences (P < 0.05).

Reviewer comment -L. 236-238. Possible reasons of the discrepancy between α-amylase inhibition activity and antioxidant properties of the samples should be exposed and discussed.

Respond to Reviewer 1: Interaction between plant polyphenols and a-amylase activity inhibition has become the subject of recent interest in postprandial hyperglycaemia [25]. Accordingly, consumption of starch largely determines postprandial blood sugar levels, and also affects glucose metabolism [25]. Postprandial hyperglycaemia has been implicated in the disturbance of carbohydrate metabolism. Delaying any increase in blood glucose levels is therefore regarded as useful to the mitigation of insulin resistance and/or type II diabetes. Starch is largely digested to reducing sugars (such as maltose, maltotriose and amylodextrin) by α-amylase in the small bowel. These reducing sugars are subsequently further hydrolyzed by α-glucosidase, resulting in glucose. α-Amylase is therefore a particularly important enzyme in starch hydrolysis. Studies have recommended that enzyme activity be regulated by both chemical and biological components in order to prevent and treat postprandial hyperglycaemia and associated metabolism disorder [24, 25]. There is a very close association between the inhibitory activity of a polyphenol against α-amylase and the phenolic molecular structure, and the relationships between structure and inhibition have been the subject of previous investigation [25]. In terms of flavonoids, in particular, the presence of hydroxyls (-OH) at the 5-, 6-, and 7-positions of ring A and at the 4′-position of ring B is capable of increasing the inhibitory activity due to the important role played by –OH in the formation of hydrogen bonds with the enzyme’s active site [45]. The conjugation of 4-carbonyl with 2, 3-double bonds also makes a significant contribution to flavonoids’ inhibitory properties. This is principally due to this conjugation heightening electron delocalization between A and C rings, thus enhancing the stability of π-π stacking between the flavonoid aromatic rings and the indole ring of tryptophan at the active site of α-amylase [25, 45]. Moreover, galloyl moiety has recently been proposed as an essential substitution for α-amylase inhibition by tea polyphenols and gallotannins [25, 26]. This is attributable to the relatively powerful non-covalent interactions taking place between the moiety and the enzyme, including hydrogen bondings between –OH of galloyl and the catalytic amino acid residues (e.g. Glu233), and hydrophobic π-π conjugation (aromatic-aromatic) between the galloyl benzene ring and tryptophan aromatic rings at the enzyme active site [25]. It is generally agreed that the antioxidant activity of phenolic compounds is often related to the chemical composition of individual compounds, which is dependent on a variety of factors, such as geographic variation, harvest time, environmental and agronomic conditions, the botanical parts of plants, and extraction methods [46]. Consistent with this, the difference in a-amylase acitivity inhibitions in the eggplants can be attributed to the above-cited chemical (different phenolic molecular structure, antocyanin or non-anthocyanin phenolics, pH, PPO acticity, etc.) and geographical factors. 

Reviewer comment -L. 267-270  Results of the PCA are not surprising, since peel and fruits are very different in all the variables measured. Moreover, nothing is said about how this analysis has been made. In addition of the results concerning PCA, the meaning of this analysis must be discussed. I mean, which are the reasons of doing such analysis? What essential information does it provide? What are the main conclusions?

Respond to Reviewer 1: PCA is the most popular multivariate statistical analysis used by almost all scientific disciplines. It analyzes a data table representing observations described by several dependent variables, which are, in general, inter-correlated. Its aim is to extract the important information from the data table and to express this information as a set of new orthogonal variables known as principal components. It also represents patterns of similarity of observations and variables by displaying them as points in maps [47] that can summarize the dimensionality of high-dimensional complex data through a smaller set of “summary indices” that can be easly visualized and analyzed. In recent years, researchers have used this method of analysis to determine whether any of the observations or variables differ significantly among treatments [48, 49, 50].

Reviewer comment -L. It seems that a total of 24 eggplants were used for each cultivar. Were the plants pooled before analysis? Please explain. 

Respond to Reviewer 1: …. Plant or animal debris were immediately removed from the eggplants fruits, which were washed in double-distilled water, kept below 5 °C and transported within approximately 3 h in a portable cold storage box. At the laboratory, peel and pulp samples were prepared in line with the sampling protocol described by Stommel and Whitaker [4] for eggplants, with slight modifications. In brief, the fruits were peeled using a porcelain fruit/vegetable peeler within one hour. A 2.5 cm longitudinal

Reviewer comment -L. 355. In my opinion, the term “whole fruit” in this context is incorrect, since peel has been previously removed (according with the description). Therefore, it would be more appropriate to call it "pulp", to avoid confusions.

Respond to Reviewer 1: ‘whole fruit’ was replaced with ‘pulp’ throughout the text.

Reviewer comment -L. 357. Include the negative sign before number 80.

Respond to Reviewer 1: …. and stored at - 80°C…

Reviewer comment -L- 363. One extraction followed by triple extraction make a total of four extractions… were there 3 or 4 extractions in total?

-L. 364. Replace letter x by symbol × (also L. 377 and others). There is no need of indicate speed in g and in rpm. Use rpm only when characteristics of the motor and centrifuge are given. The same in other parts of the text.

-L. 369. Check subsection number. Which aqueous extract? samples are supposed to be lyophilized. Please explain.

-L. 377. What means “3 x 3 mL deionized water”?

Respond to Reviewer 1:

3.3. Extraction of Phenolics

Crude phenolic extracts (CE) of peel and pulp were prepared by modifying the method described by Rodriguez-Saona and Wrolstad [51]. All extractions were performed in triplicate. Approximately 3 g of eggplant sample prepared as described above was extracted using 50 mL of 80% aqueous methanol (80:20, methanol:water, v/v) , followed by triple extraction using the same solvent until getting a clear supernatant was obtained. The homogenates were combined and centrifuged at 1500 rpm in a M2 rotor (Hermle Z 326 K, Hermle Labortechnik, Wehingen, De, Germany) for 30 min at 4 °C. The supernatants were concentrated using a rotary evaporator (Laborata 4003, Heidolph Instruments, Schwabach, Germany) at 38 °C. The slury was dried using a freeze-dryer and dissolved in 10 mL deionized water (aquesous extract) for further analysis.  

Next, the aqueous extract was fractioned by solid-phase extraction (SPE) using Thermo HyperSep™ C18 cartridges (max 500 mg packed bed, 3 mL, Waltham, MA USA) to obtain subextracts (fractions). The extraction columns were rinsed with 80% methanol (9 mL) and then activated using deionized water (9 mL)  followed by triple wash. The aqueous sample was then passed through the columns.

Reviewer comment -L. 387-401. In this paragraph the determination of anthocyanin content is lacking.

Respond to Reviewer 1: Total anthocyanin contents (ACY) of the eggplant fruit extracts were estimated spectrophotometrically according to Giusti et al. [54]. The anthocyanin content was calculated using the equation; anthocyanin content (g kg −1 fw) = A × MW × DF/(ϵ × W); where: A = absorbance (A520 nm − A700 nm)pH 1.0 − (A520 nm − A700 nm)pH 4.5, MW = molecular weight of delphinidin-3-O-glucoside (C21H21O12, 465.4, del-3-glc), DF = dilution factor, ϵ = molar absorptivity (27 000 M-1 cm-1), and W = sample weight (kg). The results were expressed as g del-3-glc kg-1 on a fresh-weight basis (fw).

Reviewer comment -L. 441 and 478. Check degree symbol.

Respond to Reviewer 1:

3.6. UHPLC-MS/MS Determination of Phenolic acids in Eggplants: Phenolic acids of peel and pulp of each eggplant cultivar were fractionated as free, esterified, glycosided and ester-bounded phenolic acids using previously described methods [57, 58]. One gram of the pulverized and dried peel or pulp of eggplant samples was first extracted using aqueous methanol (80:20, v/v, 3 x 20 mL) including 2,6-di-tert-butyl-p-cresol (DBC; 6 mg/100 mL). The extraction was performed in triplicate until the solution became colourless. The combined homogenates were centrifuged at 6000 rpm for 15 min at 4 °C. The supernatants

Reviewer comment -L. 437-470. Sections 3.7 and 3.8 may be joined into just one.

Respond to Reviewer 1:

3.6. UHPLC-MS/MS Determination of Phenolic acids in Eggplants

Phenolic acids of peel and pulp of each eggplant cultivar were fractionated as free, esterified, glycosided and ester-bounded phenolic acids using previously described methods [57, 58]. One gram of the pulverized and dried peel or pulp of eggplant samples was first extracted using aqueous methanol (80:20, v/v, 3 x 20 mL) including 2,6-di-tert-butyl-p-cresol (DBC; 6 mg/100 mL). The extraction was performed in triplicate until the solution became colourless. The combined homogenates were centrifuged at 6000 rpm for 15 min at 4 °C. The supernatants were concentrated in the rotary evaporator under reduced pressure at 35 °C. The slurry was freeze-dried in the lyophilizer, and re-dissolved in water acidified to pH 2 with HCl. Free phenolic acids were extracted into diethylether. The remaining aqueous phase was split into two parts, hydrolyzed by either 2 mol/L NaOH or 6 mol/L HCl, and extracted with diethylether after adjustment to pH 2. The sediment was hydrolyzed by either 2 mol/L NaOH or 6 mol/L HCl, and extracted with diethylether after adjustment to pH 2. Analytes were quantified using deuterium-labeled internal standards of 4-hydroxybenzoic (2,3,5,6-D4) and salicylic (3,4,5,6-D4) acids, as described previously [57, 58], with some slight modifications.

An ACQUITY Ultra Performance LC™ system (Waters, Milford, MA, USA) linked to a Micromass Quattro micro™ API benchtop triple quadrupole mass spectrometer (Waters MS Technologies, Manchester, UK) was used for analysis. Samples were injected onto the BEH C8 reversed-phase column (1.7 µm, 2.1 × 150 mm, Waters, Milford, MA). Phenolic acids were identified and quantified according to Gruz et al. [58]. Deuterium labeled 4-hydroxybenzoic-D4 (2,3,5,6-D4) and salicylic-D4 acids were used as internal standards in order to quantify phenolic acids.

3.7. Extraction and HPLC-DAD/ESI-MS Analysis of Anthocyanins in Eggplants

Extraction of anthocyanins in the eggplant samples was perfromed as described Lee et al. [59]. The lyophilized fruit material (0.2 g) was extracted in triplicate with 30 mL of 1% HCI in 40% methanol by shaking in the dark for 24 h at room temperature. The acidic methanol extract was centrifuged 1200 rpm for 10 min at 4°C., and an aliquot of the supernatant was filtered through Whatman No. 2 filter paper anda  0.45 µm syringe filter.

Anthocyanin quantitative analysis was conducted following the method described by Lee et al. [59]. HPLC was performed using an Agilent 1200 series (Waldbronn, Germany) quaternary pump, an Agilent 1200 series diode array detector, a wellplate autosampler and the ChemStation software (version B.04.03). The peak area of the delphinidin-3-O-rutinoside standard solution was plotted against the concentration. The stock solution was prepared with 1% TFA (v/v) in methanol to yield a 1 mg/mL concentration. Calibration curves were prepared at six different concentrations (1, 5, 10, 20, 50 and 100 μg/mL). High linearity (r2 > 0.999) was obtained for the standard curve. A 20 μL sample of acidic methanol extract from the eggplant was injected onto an analytical reversed-phase C18 column (TOSOH ODS 120T; 150 mm x 4.6 mm, 5μm, Tosoh Corporation, Japan). The mobile phase was composed of 5% formic acid in water (eluent A) and 5% formic acid in acetonitrile (eluent B). The gradient elution conditions for HPLC-DAD were 0 min, 10% B, 20 min, 30% B and 25 min 60% B. The total running time was 37 min, and the flow rate was set at 0.7 mL/min. The column temperature was 30 °C, and anthocyanins were detected by monitoring the elution at 525 nm. HPLC-ESI-MS analysis for identification of anthocyanins was performed using an Agilent 1200 series HPLC system coupled with an Agilent 6110 Quadrupole mass spectrometer (Boeblingen, Germany) equipped with an electrospray ionisation (ESI) source mass analyser. Data acquisition and processing were performed on Chemstation LC & LC-MS software (version B.04.03). The mass spectrometer conditions were capillary voltage, 4000 V; fragmentation voltage, 150 V; drying gas temperature, 350 °C; gas flow of N2, 12 L/min; and nebuliser pressure, 50 psi. The instrument was operated in positive ion mode, scanning from m/z 100 to 1000 at a scan rate of 1.45 s/cycle.

Reviewer comment -L. 328…. 489.In the Material and Methods section, the statistical analysis is lacking.

 3.9. Statistical Analysis

All analyses were performed using a completely randomized design. Three biological replicates, each with three technical replicates each (n = 6) were performed for peel and pulp. All data were subjected to one-way analysis of variance (ANOVA) and the significance of differences in contents of compounds/chemical components was evaluated using Duncan’s Multiple Range Test with a significance threshold of P < 0.05. A statistical software package was also used to perform principal components analysis (PCA) (Addinsoft 2019, XLSTAT and Data Analysis Solution, version 2019.3.2., New York, NY, USA). The correlation coefficients (r) were determined for the phenolic contents (total phenolic contents, phenolic acids and anthocyanins, etc.) and antioxidant capacity values levels comparing mean peel and pulp values.

Reviewer 2 Report

The study is well designed and methods used are appropriate. However some minor changes should be performed as suggested:

  • paragraph 2.1 of Results and Discussion section: introduce in the text all abbreviations used, except for ORAC and DPPH.
  • paragraph 2.2 lines 179-189: why cholorogenic acid data were presented in a separate table (table 4) and not in table 2 together with other phenolic acids?  Moreover the concentration of chlorogenic acid was expressed in microgram/kg in table 4, while data reported in literature by the authors are in the order of thousand milligram/kg (line 192-193. Please verify and check the units used throughout the manuscript.
  • figure 2 please specify in the legend of the figure the HPLC used (HPLC-DAD). The peaks 1, 2 and 3 in the chromatograms are almost undetectable. Please enlarge the chromatograms.
  • in table 3 the peak 1,2 and 3 corresponding to del-3-rut-5-gluc, del-3-rut-glc and del-3-glc respectively were not detected by HPLC-MS. Why the same compounds were detected in figure 2 with HPLC-DAD? Please explain.
  • paragraph 3.2 of Materials and Method section: line 357 please add minus before 80;
  • paragraph 3.3 of Materials and Methods section: line 367 "at below 38°C", specify if the temperature was at 38°C or below (or ≤ 38°C). line 369, "next, 10 ml of the aqueous extract" please specify at what kind of aqueous extract you are referring to. How this extract has been obtained? lines 383-385, how the methanolic residue was analyzed? please describe.
  • paragraph 3.6 of Materials and Methods section: lines 426-428, please describe shortly the method used (Ayaz et al.) to obtain the different fractions (free, esters, glycoside, ester-bound) of phenolic acids.
  • paragraph 4. Conclusion lines 496-498: specify the colour "The peel of black and purple eggplants...."

Author Response

Comments of Reviewer 2

  • paragraph 2.1 of Results and Discussion section: introduce in the text all abbreviations used, except for ORAC and DPPH.
  •  

Respond to Reviewer 2: All abbreviations were included where they first reported.

  • paragraph 2.2 lines 179-189: why cholorogenic acid data were presented in a separate table (table 4) and not in table 2 together with other phenolic acids?  Moreover the concentration of chlorogenic acid was expressed in microgram/kg in table 4, while data reported in literature by the authors are in the order of thousand milligram/kg (line 192-193. Please verify and check the units used throughout the manuscript.

Respond to Reviewer 2: The most widely distributed phenolic acid ester in eggplants is chlorogenic acid (CGA, 5-caffeoylquinic acid), and these vegetables are highly prized in the human diet because of this high content [3, 18, 32]. In the present study, CGA was also detected in four forms (Table 3), its concentration (mg/kg fw) in peel being significantly (P < 0.05) higher than that in pulp. For instance, CGA occurred at high concentrations in free form in the peel in ‘Kadife Kemer’ (27.55), in the peel of the black eggplant (Aydin Siyahi) in ester (7.82), glycoside (294.1) and ester-bound (2.41) forms. However, the pulp contained the highest CGA concentration in free (4.71), ester (3.06) and ester-bound (1.23) forms in ‘Aydin Siyahi’, and in the glycoside form (60.12) in ‘Kadife Kemer’. Our findings are in good agreement with those of other authors who reported a broad variation in CGA concentration in eggplants, ranging from 8600 to 17000 mg/kg fw [3, 32].

Table 3. Chlorogenic acid contents (mg/kg fresh weight) in fruits in common eggplants*. 

‘Aydin Siyahi’, black

‘Kadife Kemer’, purple

‘Trabzon Kadife’, white

Chlorogenic acid (CGA)

Peel

Pulp

Peel

Pulp

Peel

Pulp

Free form (F)

6.08 ± 0.44 D**

4.71 ± 0.10 C

27.55 ± 0.82 E

1.14 ± 0.11 B

0.69 ± 0.03 AB

0.22 ± 0.02 A

Ester form (E)

7.82 ± 0.36 E

3.06 ± 0.23 D

0.46 ± 0.09 BC

0.00 ± 0.00 A

0.54 ± 0.07 C

0.15 ± 0.00 AB

Glycoside form (G)

294.05 ± 4.81 F

52.28 ± 0.94 B

108.16 ± 1.26 D

60.12 ± 0.20 C

213.45 ± 2.67 E

15.68 ± 0.54 A

Ester-bound form (EB)

2.41 ± 0.23 E

1.23 ± 0.04 D

1.00 ± 0.01 C

0.29 ± 0.02 B

0.43 ± 0.01 B

0.10 ± 0.00 A

* Values represent mean ± SD of three independent experiments extractions and determinations. 

** different capital letters differ significantly at P < 0.05 (one-way ANOVA was used for comparisons).

  • figure 2 please specify in the legend of the figure the HPLC used (HPLC-DAD). The peaks 1, 2 and 3 in the chromatograms are almost undetectable. Please enlarge the chromatograms.

Respond to Reviewer 2: The figure was revised in light of the comment.

Figure 1. HPLC-DAD chromatograms of anthocyanins (l = 525 nm) in peel and pulp fruit in eggplant fruits (Aydin Siyahi, Kadife Kemer and Trabzon Kadife).  Del-3-rut-5-glc; delphinidin-3-rutinoside-5 glycoside, Del-3-rut-glc; delphinidin-3-rutinoside-glycoside, Del-3-glc; delphinidin-3-glycoside, Del-3-rut; delphinidin-3-rutinoside.

  • in table 3 the peak 1,2 and 3 corresponding to del-3-rut-5-gluc, del-3-rut-glc and del-3-glc respectively were not detected by HPLC-MS. Why the same compounds were detected in figure 2 with HPLC-DAD? Please explain.

Respond to Reviewer 2: Table 4 was revised in light of the comment

Table 4. HPLC chromatographic characteristics and delphinidin-3-O-rutinoside (del-3-rut) concentrations (mg/kg fw) in eggplants using HPLC-DAD/ESI-MSc.

Anthocyaninf

Peak No

Retention Time

(Rt, min)

λmax

(nm)

MS

[M + H]+

‘Aydin Siyahi’, black

‘Kadife Kemer’, purple

‘Trabzon Kadife’, white

peel

pulp

peel

pulp

peel

pulp

del-3-rut-5-glc

1

5.9

525

773.1

trace j

trace

trace

trace

trace

trace

del-3-rut-glc

2

7.6

522

773.1

trace

trace

trace

trace

trace

trace

del-3-glc

3

8.9

528

465.1

trace

trace

trace

trace

trace

trace

del-3-rut

4

9.7

526

611.1

1162.22 ± 5.56 D*

194.62 ± 4.54 B

336.59 ± 11.20 C

45.45 ± 1.70 A

215.11 ± 2.9 2 B

72.44 ± 1.97 A

c Values represent mean ± SD of three independent experiments extractions and determinations. 

f del-3-rut-5-glc; delphinidin-3-rutinoside-5 glycoside, del-3-rut-glc; delphinidin-3-rutinoside-glycoside, del-3-glc; delphinidin-3-glycoside, del-3-rut; delphinidin-3-rutinoside.

j trace; trace anthocyanin, not quantified

* different capital letters in each row differ significantly at P < 0.05 (one-way ANOVA was used for comparisons).                                                                                                                  

  • paragraph 3.2 of Materials and Method section: line 357 please add minus before 80;

Respond to Reviewer 2: The dried peel and pulp samples from each eggplant were then pulverised using an agate mortar and pestle and stored at - 80°C for further analyses.

  • paragraph 3.3 of Materials and Methods section: line 367 "at below 38°C", specify if the temperature was at 38°C or below (or ≤ 38°C). line 369, "next, 10 ml of the aqueous extract" please specify at what kind of aqueous extract you are referring to. How this extract has been obtained? lines 383-385, how the methanolic residue was analyzed? please describe.

Respond to Reviewer 2:….

3.3. Extraction of Phenolics

Crude phenolic extracts (CE) of peel and pulp were prepared by modifying the method described by Rodriguez-Saona and Wrolstad [51]. All extractions were performed in triplicate. Approximately 3 g of eggplant sample prepared as described above was extracted using 50 mL of 80% aqueous methanol (80:20, methanol:water, v/v) , followed by triple extraction using the same solvent until getting a clear supernatant was obtained. The homogenates were combined and centrifuged at 1500 rpm in a M2 rotor (Hermle Z 326 K, Hermle Labortechnik, Wehingen, De, Germany) for 30 min at 4 °C. The supernatants were concentrated using a rotary evaporator (Laborata 4003, Heidolph Instruments, Schwabach, Germany) at 38 °C. The slury was dried using a freeze-dryer and dissolved in 10 mL deionized water (aquesous extract) for further analysis.  

Next, the aqueous extract was fractioned by solid-phase extraction (SPE) using Thermo HyperSep™ C18 cartridges (max 500 mg packed bed, 3 mL, Waltham, MA USA) to obtain subextracts (fractions). The extraction columns were rinsed with 80% methanol (9 mL) and then activated using deionized water (9 mL)  followed by triple wash. The aqueous sample was then passed through the columns.

  • paragraph 3.6 of Materials and Methods section: lines 426-428, please describe shortly the method used (Ayaz et al.) to obtain the different fractions (free, esters, glycoside, ester-bound) of phenolic acids.

Respond to Reviewer 2: … The slurry was freeze-dried in the lyophilizer, and re-dissolved in water acidified to pH 2 with HCl. Free phenolic acids were extracted into diethylether. The remaining aqueous phase was split into two parts, hydrolyzed by either 2 mol/L NaOH or 6 mol/L HCl, and extracted with diethylether after adjustment to pH 2. The sediment was hydrolyzed by either 2 mol/L NaOH or 6 mol/L HCl, and extracted with diethylether after adjustment to pH 2. Analytes were quantified using deuterium-labeled internal standards of 4-hydroxybenzoic (2,3,5,6-D4) and salicylic (3,4,5,6-D4) acids, as described previously [57, 58], with some slight modifications….

  • paragraph 4. Conclusion lines 496-498: specify the colour "The peel of black and purple eggplants...."

Respond to Reviewer 2:  It was corrected.

Reviewer 3 Report

This is a very interesting and detailed study on the phenolic composition of 3 eggplant cultivars grown in Turkey. Both methodology and findings are well-documented, figures and tables provide useful information, and authors provide sufficient references to compare and support their results. Some minor comments:

  1. Line 357 “…and stored at 80°C for further analyses.” I suppose it is -80, please correct
  2. In 3.4 Determination of TPC, line 392, authors state “After cooling at room temperature…” but previously they have not mentioned if the reaction mixture was heated. Please define
  3. Line 441 “…for 30 min at 4oC., and…”, please correct the degree sign

Author Response

Comments of Reviewer 3

  1. Line 357 “…and stored at 80°C for further analyses.” I suppose it is -80, please correct

Respond: It was corrected as follow; ……mortar and pestle and stored at - 80°C for further analyses.

  1. In 3.4 Determination of TPC, line 392, authors state “After cooling at room temperature…” but previously they have not mentioned if the reaction mixture was heated. Please define

Respond: Normally no heating in the reaction mixture. As the sample, Folin-Ciocalteu reagent and

Na2CO3 mixed, the reaction mixture has its own reaction temperature around at 30°. Therefore, avoiding any abnormal absorbance reading, the reaction mixture always are awaited to cool to reach at 25°. It is also confirmed with the reference (Slinkard, K.; Singleton, V.L. Total phenol analysis: automation and comparison with manual methods. Am. J. Enol. Vitic. 1977, 28, 49-55).

  1. Line 441 “…for 30 min at 4oC., and…”, please correct the degree sign

Respond: It was corrected as follow;

……………4,427 x g (6000 rpm) for 30 min at 4°C., and…………….

Round 2

Reviewer 1 Report

The manuscript has been extensively reviewed according to the suggestions and it has greatly improved. The paper is ready to be published. 

Reviewer 2 Report

none